# ACE2 pathway regulates thermogenesis and energy metabolism

**Xi Cao[1†], Ting-Ting Shi[1†], Chuan-Hai Zhang[2†], Wan-Zhu Jin[3], Li-Ni Song[1], Yi-Chen Zhang[1], Jing-Yi Liu[1], Fang-Yuan Yang[1], Charles N Rotimi[4], Aimin Xu[5], Jin-Kui Yang[1]\***

[1]Beijing Key Laboratory of Diabetes Research and Care, Department of Endocrinology, Beijing Diabetes Institute, Beijing Tongren Hospital, Capital Medical University, Beijing, China; [2]Department of Physiology, UT Southwestern Medical Center at Dallas, Dallas, United States; [3]Key Laboratory of Animal Ecology and Conservation Biology, Institute of Zoology, and State Key Laboratory of Brain and Cognitive Sciences, Institute of Biophysics, Chinese Academy of Sciences, Beijing, China; [4]Center for Research on Genomics and Global Health, National Human Genome Research Institute, National Institutes of Health, Bethesda, United States; [5]State Key Laboratory of Pharmaceutical Biotechnology, Department of Medicine, University of Hong Kong, Hong Kong, China

**\*For correspondence:**
jkyang@ccmu.edu.cn

[†]These authors contributed equally to this work

**Competing interest:** The authors declare that no competing interests exist.

**Abstract** Identification of key regulators of energy homeostasis holds important therapeutic promise for metabolic disorders, such as obesity and diabetes. ACE2 cleaves angiotensin II (Ang II) to generate Ang-(1-7) which acts mainly through the Mas1 receptor. Here, we identify ACE2 pathway as a critical regulator in the maintenance of thermogenesis and energy expenditure. We found that ACE2 is highly expressed in brown adipose tissue (BAT) and that cold stimulation increases ACE2 and Ang-(1-7) levels in BAT and serum. *Ace2* knockout mice (*Ace2*[-/y]) and *Mas1* knockout mice (*Mas1*[-/-]) displayed impaired thermogenesis. Mice transplanted with brown adipose tissue from *Mas1*[-/-] display metabolic abnormalities consistent with those seen in the *Ace2* and *Mas1* knockout mice. In contrast, impaired thermogenesis of *Lepr*[db/db] obese diabetic mice and high-fat diet-induced obese mice were ameliorated by overexpression of *Ace2* or continuous infusion of Ang-(1-7). Activation of ACE2 pathway was associated with improvement of metabolic parameters, including blood glucose, lipids, and energy expenditure in multiple animal models. Consistently, ACE2 pathway remarkably enhanced the browning of white adipose tissue. Mechanistically, we showed that ACE2 pathway activated Akt/FoxO1 and PKA pathway, leading to induction of UCP1 and activation of mitochondrial function. Our data propose that adaptive thermogenesis requires regulation of ACE2 pathway and highlight novel potential therapeutic targets for the treatment of metabolic disorders.

## Editor's evaluation

Authors have found that ACE2 is highly expressed in brown adipose tissue (BAT), indicating ACE2 pathway as a critical regulator in the maintenance of thermogenesis and energy expenditure. Identifying new regulators of energy homeostasis authors have shed light on novel potential therapeutic targets for the treatment of metabolic disorders.

## Introduction

Energy imbalance and the associated metabolic syndromes have become a worldwide public health problem. Thus, identifying factors that can stimulate energy expenditure is instrumental to the development of therapeutics to reduce obesity associated disorders that affect over 10% of the world

population (*Dong et al., 2018*). In the renin-angiotensin system (RAS), angiotensin-converting enzyme 2 (ACE2) cleaves angiotensin II (Ang II) to generate angiotensin-(1-7) (Ang-(1-7)). Ang-(1-7) is a hepta-peptide hormone which acts mainly through G-protein-coupled receptor Mas1 (*Santos et al., 2003*). ACE2-Ang-(1-7)-Mas1 pathway works as a negative regulator of ACE-Ang II pathway in multiple disease states (*Clarke and Turner, 2012*).

Our group focused on ACE2 originated from concerns of severe acute respiratory syndrome (SARS) in 2003. Ambient hyperglycemia occurred very early in SARS patients and was an independent predictor for death and morbidity in SARS patients (*Yang et al., 2006*). Interestingly, ACE2 is the functional receptor for SARS coronavirus (*Lavoie and Sigmund, 2003*). We reported that *Ace2* knockout (*Ace2$^{-/y}$*) mice exhibited progressive impairments in glucose tolerance indicating that ACE2 is a potential new target for the treatment of type 2 diabetes (*Niu et al., 2008*). In addition, we have demonstrated that ACE2 regulates mitochondrial function in pancreatic β-cells, inhibits hepatic insulin resistance, ameliorates hepatic steatosis, and improves glucose uptake in adipocytes (*Cao et al., 2019*; *Cao et al., 2016*; *Cao et al., 2014*; *Liu et al., 2012a*; *Song et al., 2020*; *Yang et al., 2018*). These findings support the hypothesis that the ACE2-Ang-(1-7)-Mas1 axis may have protective effects on metabolic syndrome.

In this study, we reported the effects of ACE2 pathway on regulating thermogenesis and energy metabolism via modulating mitochondrial function. We found that *Ace2$^{-/y}$* and *Mas1* knockout (*Mas1$^{-/-}$*) mice are cold intolerance. We provided compelling genetic, metabolic, physiological, histological, cellular, and molecular evidence to demonstrate that ACE2 pathway is a critical regulator in the maintenance of energy expenditure. This pathway regulates function of brown adipose tissue (BAT) and systemic energy metabolism. Mechanistically, ACE2 pathway activates both Akt/FoxO1 signaling and PKA signaling, leading to induction of uncoupling protein-1 (UCP1) and activation of mitochondrial function. Therefore, ACE2 pathway is a potential treatment target for metabolic disorders including diabetes, obesity, and even cardiovascular diseases.

## Results

### Acute cold exposure increases components of ACE2 pathway

The major tissue of the body where energy is converted into the form of heat to maintain the body temperature is BAT. We found both mRNA level and protein level of *Ace2* and *Mas1* in BAT were obviously higher than the ones in subcutaneous white adipose tissue (scWAT) and epididymal white adipose tissue (eWAT) in mice (*Figure 1A and B*). Acute cold exposure caused a significant up-regulation of ACE2 protein expression in BAT (*Figure 1C*). Meanwhile, *Ace2* mRNA levels in BAT, scWAT and eWAT, and *Mas1* mRNA levels in BAT and eWAT were increased after exposed to 4 °C for 48 hr (*Figure 1D and E*). ACE2 and Ang-(1-7) were also marginally increased in serum upon cold challenge (*Figure 1F and G*). These results demonstrated a selective induction of ACE2 pathway in thermogenic adipose depots (BAT and scWAT) in response to cold environment.

### ACE2 promotes thermogenesis and energy metabolism

To explore the physiological roles of ACE2 in cold-induced adaptive thermogenesis, we used the HFD-induced *Ace2$^{-/y}$* mice. ACE2 is essential for expression of neutral amino acid transporters in the gut (*Hashimoto et al., 2012*). This is consistent with our observation that *Ace2$^{-/y}$* mice fed an HFD displayed significantly decreased weight compared to wild-type (WT) mice (*Figure 2A*). Serum Ang-(1-7) levels were decreased in the *Ace2$^{-/y}$* mice (*Figure 2B*). Consistent with previous studies (*Cao et al., 2014*; *Liu et al., 2012a*; *Niu et al., 2008*; *Shi et al., 2018*; *Zhang et al., 2016*), *Ace2$^{-/y}$* mice had an impaired glucose tolerance and abnormal lipid profiles (*Figure 2—figure supplement 1A-C*).

A key factor for controlling energy homoeostasis is the balance between caloric intake and energy expenditure. Thus, we measured energy expenditure using a comprehensive laboratory animal monitoring system (CLAMS). We observed a decreased oxygen consumption (VO$_2$), carbon dioxide release (VCO$_2$) and energy expenditure (EE) in *Ace2$^{-/y}$* mice (*Figure 2C, D and E*), without observable changes in food and/or water intake as well as physical activity, compared to the WT mice (*Figure 2—figure supplement 1D-F*).

To further examine the differences in energy expenditure among these animals, we performed a cold tolerance test in order to gauge adaptive thermogenesis. *Ace2$^{-/y}$* mice had lower thermogenesis

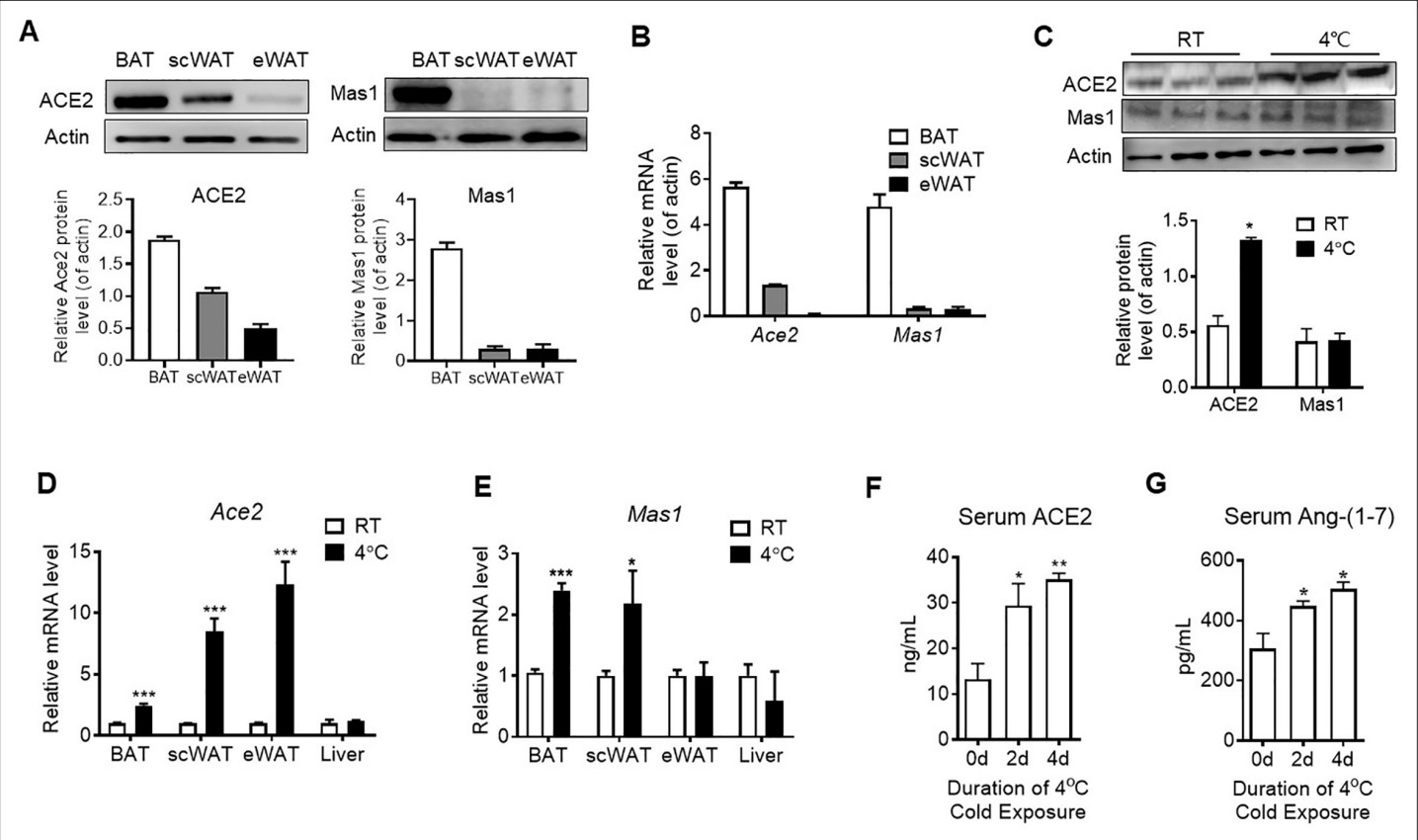

**Figure 1.** ACE2 pathway is activated by cold exposure. Eight-week-old male C57BL/6 J mice were housed at room temperature (RT) for 2 weeks before cold exposure at 4 °C for various time periods as indicated. (**A**) Levels of ACE2 and Mas1 protein from interscapular brown adipose tissue (BAT), subcutaneous and epididymal white adipose tissue (scWAT and eWAT) of C57BL/6 mice at room temperature (RT), as determined by Western blotting (n = 3/each group). (**B**) Levels of *Ace2* and *Mas1* mRNA from BAT, scWAT and eWAT of C57BL/6 mice at RT, as determined by qPCR (n = 3/each group). (**C**) Levels of ACE2 and Mas1 protein from interscapular BAT of C57BL/6 mice at RT or exposed to 4 °C for 6 hr, as determined by western blotting (n = 3/each group). (**D, E**) Levels of *Ace2* and *Mas1* mRNA from BAT, scWAT, eWAT and liver of C57BL/6 mice exposed to 4 °C for 24 hr, as determined by qPCR (n = 6/each group). (**F, G**) Serum levels of ACE2 (**F**) and Ang-(1-7) (**G**), as determined by ELISA (n = 4–6/each group). Data represent mean ± SEM. *p < 0.05, **p < 0.01 and ***p < 0.001 *vs* Control group by Student's *t*-test, or one-way ANOVA.

The online version of this article includes the following source data for figure 1:

**Source data 1.** Numerical quantification data for *Figure 1*.

than the WT mice in a cold environment (4 °C) (*Figure 2—figure supplement 1G*). To explore the source of thermogenesis, we analyzed the non-shivering thermogenesis (NST) of *Ace2^{-/y}* mice in thermoneutral condition (30 °C), ambient temperature (22 °C) and acute cold (4 °C) for 8 hr. *Ace2^{-/y}* mice had lower thermogenesis than the WT mice in either 22 °C or 4 °C (*Figure 2F*). This temperature difference was monitored by an infrared camera at 4 °C (*Figure 2G*).

To investigate whether ACE2-induced thermogenesis was related to BAT function, we performed the Positron emission tomography–computed tomography (PET-CT) analysis and the results showed a higher PET-CT signal in BAT of the HFD-induced WT mice than *Ace2^{-/y}* mice (*Figure 2H*). As expected, BAT in *Ace2^{-/y}* mice displayed larger lipid droplets but reduced multilocular structures compared to the WT mice, and reduced UCP1 expression (*Figure 2I*).

To evaluate the significance of cold-induced ACE2 for thermogenic function of BAT, the expression levels of a network of genes and proteins controlling energy expenditure and thermogenic programming were measured. Protein levels (UCP1, PGC1α, and ATP5A) (*Figure 2J*) and mRNA levels (*Ucp1, Prmd16,* and *Pparg* (*Figure 2—figure supplement 1H*)) in BAT from *Ace2^{-/y}* were obviously decreased.

To validate the above-mentioned change of thermogenesis of BAT was cell autonomous, primary brown adipocytes from *Ace2^{-/y}* mice was fractionated and differentiated in vitro. Notably, the protein and mRNA expression of known BAT markers were robustly decreased in *Ace2*-deficient primary

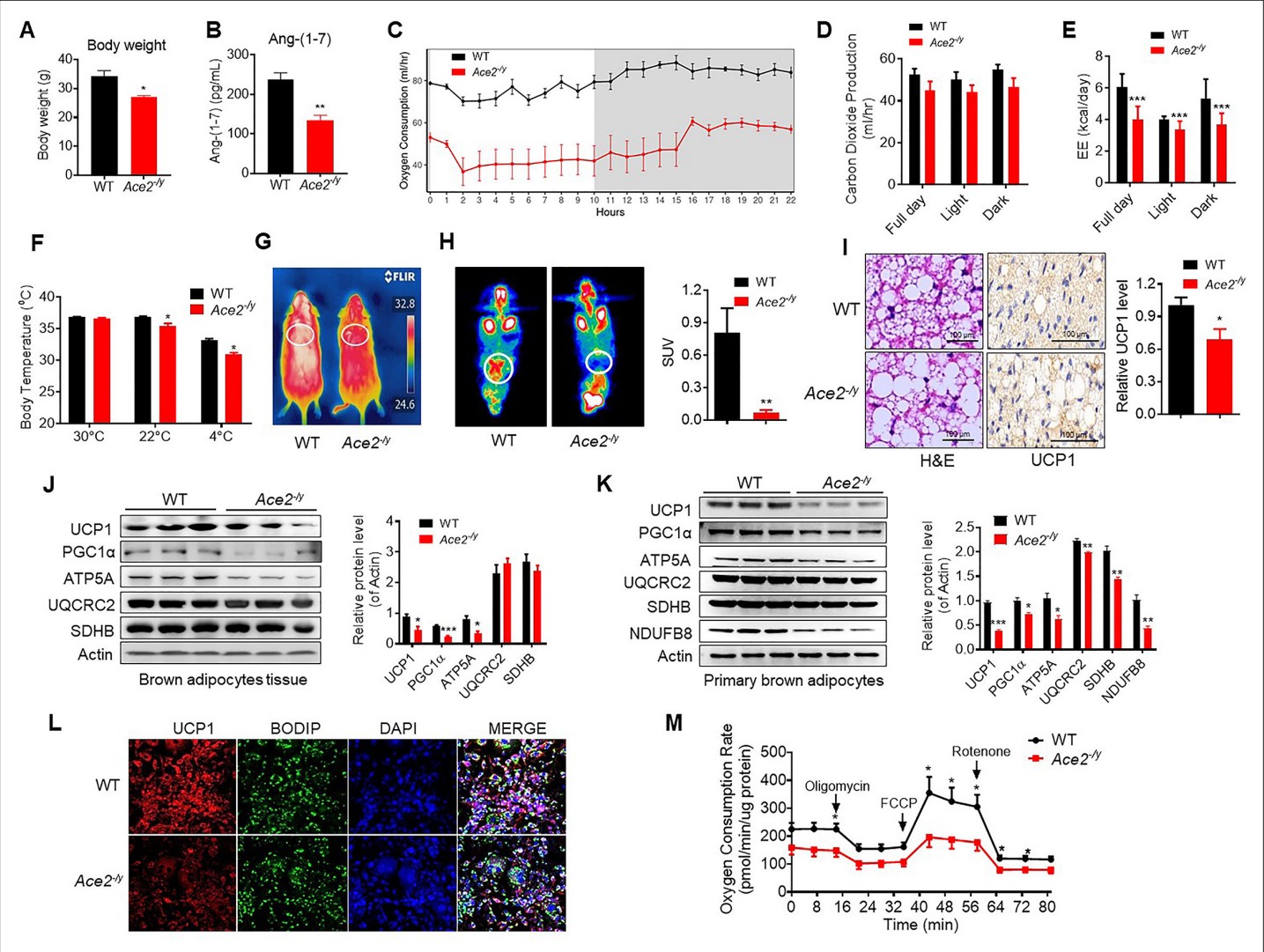

**Figure 2.** *Ace2* deficiency impairs thermogenesis, BAT activity, and energy metabolism. Eight-week-old male *Ace2⁻/y* mice and their wild-type (WT) mice (controls) had a high-fat diet (HFD) for 8 weeks. (**A**) Body weight of *Ace2⁻/y* and WT mice fed a HFD for 8 weeks (n = 4–5/each group). (**B**) Serum levels of Ang-(1-7), as determined by ELISA (n = 4–6/each group). (**C–E**) Energy expenditure was evaluated by measurement of oxygen consumption (VO₂) (**C**), carbon dioxide release (VCO₂) (**D**) and energy expenditure (EE) (**E**) over a 24 hours period (n = 5/each group). (**F**) Core body temperature at 30 °C, 22°C and 4°C for 8 hr in *Ace2⁻/y* and WT mice (n = 5/each group). (**G**) Infrared thermal images at 22 °C in *Ace2⁻/y* and WT mice. (**H**) Representative tomography–computed tomography (PET-CT) image and standard uptake values (SUVs) (n = 4/each group). (**I**) Representative haematoxylin and eosin (H&E) staining and uncoupling protein-1 (UCP1) immunostaining from BAT sections of *Ace2⁻/y* and WT mice exposure at 4 °C (n = 5/each group). (**J**) Representative western blots showing the changes of key proteins of energy expenditure and thermogenesis in BAT of *Ace2⁻/y* and WT mice exposure at 4 °C (n = 3/each group). (**K**) Representative western blots showing the key protein changes in primary brown adipocytes from *Ace2⁻/y* and WT mice (n = 3/each group). (**L**) Representative immunofluorescent images of in vitro differentiated primary brown adipocytes of *Ace2⁻/y* and WT mice, primary brown adipocytes show staining for UCP1 (red), boron-dipyrromethene (BODIPY) (green; neutral lipid dye), and DAPI (blue; nuclei). (**M**) Continuous measurement of oxygen consumption rate (OCR) in primary brown adipocytes from *Ace2⁻/y* mice and WT littermates. Oxygen consumption was performed under basal conditions, following the addition of oligomycin (1 μM), the pharmacological uncoupler FCCP (1 μM) or the Complex III and I inhibitor antimycin A and rotenone (0.5 μM) (n = 4–5/each group). Data represent mean ± SEM. *p < 0.05, **p < 0.01 *vs* WT group by Student's t-test, or ANCOVA.

The online version of this article includes the following source data and figure supplement(s) for figure 2:

**Source data 1.** Numerical quantification data for *Figure 2*.

**Figure supplement 1.** *Ace2* deficiency impairs adaptative thermogenesis by cold stimulation.

**Figure supplement 1—source data 1.** Numerical quantification data for *Figure 2—figure supplement 1*.

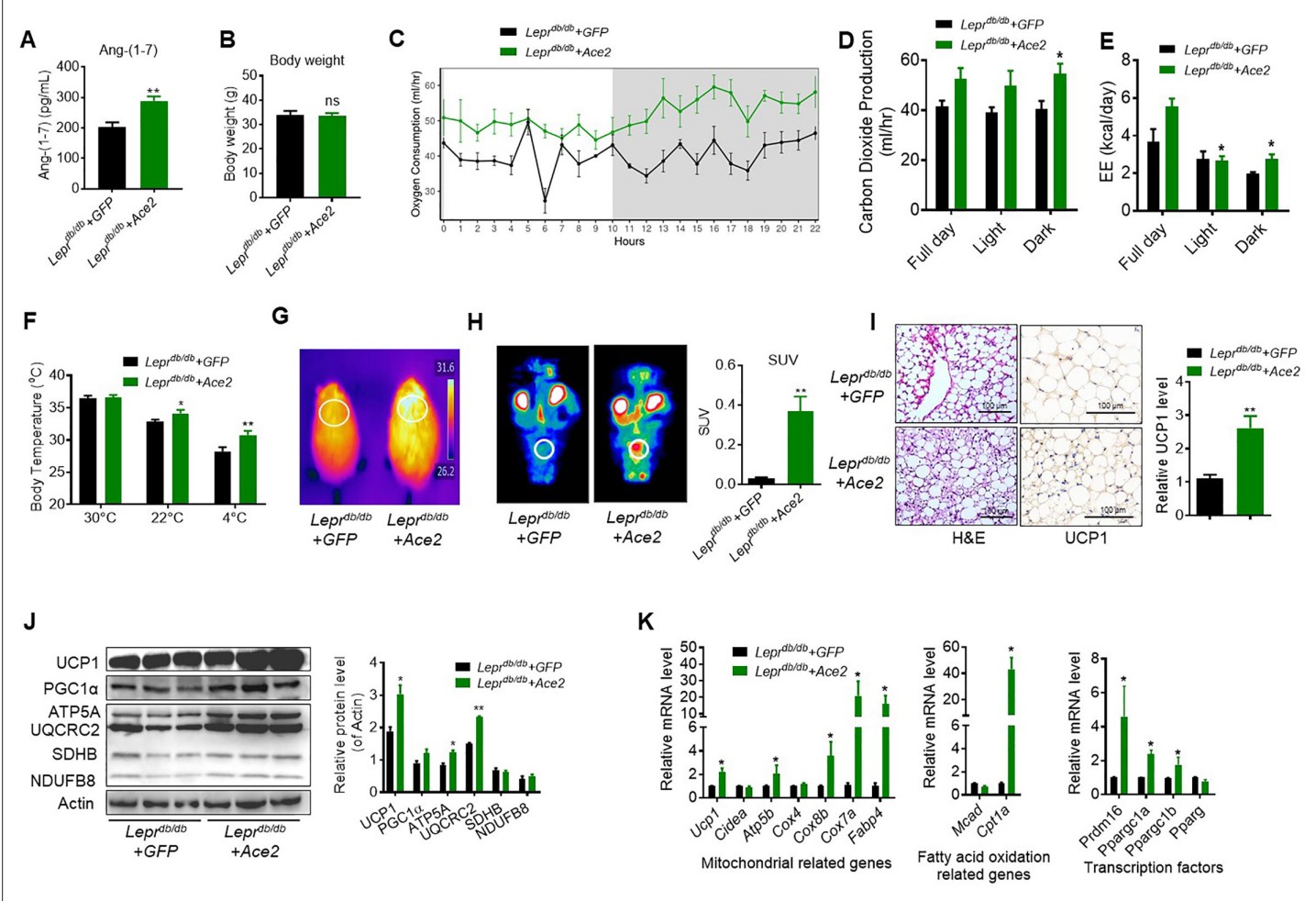

**Figure 3.** ACE2 enhances thermogenesis, BAT activity, and energy metabolism in *Lepr^db/db* obese mice. *Ace2* over-expression adenovirus (Ad-*Ace2*) and Ad-*GFP* (control) were introduced into the *Lepr^db/db* obese mice by tail vein injection. The ad-*Ace2* and Ad-*GFP* treated *Lepr^db/db* mice were used at the 6^th day post-virus injection. (**A**) Serum levels of Ang-(1-7), as determined by ELISA (n = 6–7/each group). (**B**) Body weight of ad-*Ace2* and Ad-*GFP* treated *Lepr^db/db* mice at the 6^th day post-virus injection (n = 4–6/each group). (**C–E**) Energy expenditure was evaluated by measurement of oxygen consumption (VO$_2$) (**C**), carbon dioxide release (VCO$_2$) (**D**) and energy expenditure (EE) (**E**) over a 24-hr period (n = 4–6/each group). (**F**) Core body temperature at 30 °C, 22°C and 4°C for 8 hr (n = 5/each group). (**G**) Infrared thermal images at 22 °C in *Lepr^db/db*+ *Ace2* and *Lepr^db/db*+ *GFP* mice (n = 4/each group). (**H**) Representative tomography–computed tomography (PET-CT) image and standard uptake values (SUVs) (n = 4/each group). (**I**) Representative H&E staining and UCP1 immunostaining from BAT sections of *Lepr^db/db*+ *Ace2* and *Lepr^db/db*+ GFP mice exposure at 4 °C (n = 5/each group). (**J**) Representative western blots showing the changes of key proteins of energy expenditure and thermogenesis in BAT of *Lepr^db/db*+ *Ace2* and *Lepr^db/db*+ *GFP* mice exposure at 4 °C (n = 3/each group). (**K**) Relative mRNA levels of mitochondrial related genes, fatty acid oxidation related genes and transcription factors in BAT of *Lepr^db/db*+ *Ace2* and *Lepr^db/db*+ *GFP* mice exposure at 4 °C (n = 5–6/each group). Data represent mean ± SEM. *p < 0.05, **p < 0.01 *vs* Ad-*GFP* group by Student's t-test, or ANCOVA.

The online version of this article includes the following source data and figure supplement(s) for figure 3:

**Source data 1.** Numerical quantification data for *Figure 3*.

**Figure supplement 1.** ACE2 enhance BAT activity and whole-body energy metabolism in *Lepr^db/db* and HFD mice.

**Figure supplement 1—source data 1.** Numerical quantification data for *Figure 3—figure supplement 1*.

brown adipocytes (*Figure 2K*, *Figure 2—figure supplement 1I*). Immunohistochemistry was applied to study the level of UCP1 in primary brown adipocytes differentiated from the BAT of the *Ace2^-/y* mice. The result showed the UCP1 expression was reduced in the *Ace2* deficiency primary brown adipocytes (*Figure 2L*). More importantly, the oxygen consumption rate (OCR) was significantly decreased in *Ace2*-deficient primary brown adipocytes (*Figure 2M*).

As a complementary approach to the KO mouse models, we carried out gain-of-function studies using *Ace2* over expression in obese diabetic *Lepr^db/db* mice. One week following adenovirus-induced

*Ace2* over-expression (Ad-*Ace2*) by tail vein injection in the *Lepr^db/db* mice, both ACE2 (*Figure 3—figure supplement 1A*) in BAT and circulating Ang-(1-7) (*Figure 3A*) were increased. Consistent with our previous study, the Ad-*Ace2*-treated mice exhibited an improved metabolic profile as indicated by the significant alleviation of glucose intolerance (*Figure 3—figure supplement 1B*). Notably, although no observable change on the body weight was observed in the two groups (*Figure 3B*), serum triglyceride levels decreased in the Ad-*Ace2*-treated mice (*Figure 3—figure supplement 1C*), as well as a minor change in serum cholesterol levels (*Figure 3—figure supplement 1D*).

Notably, the Ad-*Ace2*-treated *Lepr^db/db* mice had increased energy expenditure ($VO_2$, $VCO_2$ and EE) (*Figure 3C–E*). There was no obvious change in food and/or water intake as well as physical activity (*Figure 3—figure supplement 1E-G*).

We measured rectal temperature and infrared thermal imaging in the *Lepr^db/db* mice that BAT activity was defective as same as the ones in previous observations (*Trayhurn and Wusteman, 1990*; *Zhang et al., 2014*). The results showed that the thermogenesis of the *Lepr^db/db* mice was severely impaired (*Figure 3—figure supplement 1H, I*). As expected, the Ad-*Ace2* treated *Lepr^db/db* mice and HFD-induced obese mice exhibited better thermogenesis than the control mice in ambient temperature (22 °C) and acute cold (4 °C) conditions (*Figure 3F and G*, *Figure 3—figure supplement 1J,K*). Accordingly, PET-CT result showed that BAT was activated in the Ad-*Ace2*-treated *Lepr^db/db* mice (*Figure 3H*). Moreover, BAT in the Ad-*Ace2*-treated *Lepr^db/db* mice had smaller lipid droplets but increased multi-locular structures, and had increased UCP1 expression compared with the control group (*Figure 3I*).

The protein levels of UCP1, ATP5A and UQCRC2 were significantly increased in the BAT from the Ad-*Ace2* treated mice (*Figure 3J*). Consistently, the mRNA levels, including *Ucp1*, *ATP synthase F1 subunit beta* (*Atp5b*), *Cox8b*, *Cox7a*, *fatty acid binding protein 4* (*Fabp4*), *Cpt1a*, *Prdm16*, *Ppargc1a* and *Ppargc1b*, were increased in the BAT from the Ad-*Ace2*-treated *Lepr^db/db* mice (*Figure 3K*). Taken together, these results indicated that ACE2 effectively regulated the mitochondrial biogenesis and respiratory function in brown adipocytes.

## Ang-(1-7) promotes thermogenesis and energy metabolism

To explore the direct physiological roles of Ang-(1-7) in cold-induced adaptive thermogenesis, Ang-(1-7) administration by subcutaneous implantation of micro-osmotic pumps in the *Lepr^db/db* and the HFD-induced obese mice were employed. Serum Ang-(1-7) was increased in Ang-(1-7)-treated mice (*Figure 4—figure supplement 1A*). There are no significant differences in body weight between the Ang-(1-7)-treated *Lepr^db/db* mice and the *Lepr^db/db* control mice (*Figure 4A*); however, Ang-(1-7)-treated *Lepr^db/db* mice has an improved glucose tolerance ability (*Figure 4—figure supplement 1B*) and better lipid profiles (*Figure 4—figure supplement 1C, D*). The Ang-(1-7)-treated HFD-induced obese mice displayed a lower body weight compared to the control (*Figure 4B*).

Notably, the Ang-(1-7) treated *Lepr^db/db* mice had increased energy expenditure ($VO_2$, $VCO_2$, and EE) (*Figure 4C–E*) without any changes in food and/or water intake as well as physical activity (*Figure 4—figure supplement 1E-G*). Moreover, the Ang-(1-7) treated *Lepr^db/db* and the HFD-induced obese mice were better able to defend their body temperature during environmental cold (22 °C) and acute cold stress (4 °C) compared to the control (*Figure 4F–H*, *Figure 4—figure supplement 1H*). Meanwhile, the Ang-(1-7)-treated *Lepr^db/db* mice had increased multi-locular structures but smaller lipid droplets, and increased UCP1 expression comparing to the control group (*Figure 4I*). Accordingly, the Ang-(1-7)-treated *Lepr^db/db* and the HFD-induced obese mice showed more [18]F-FDG uptake in the BAT than the control mice recorded by PET-CT (*Figure 4J and K*).

The protein levels of UCP1 and PGC1α were significantly induced in the BAT from the Ang-(1-7) treated *Lepr^db/db* and the HFD-induced obese mice (*Figure 4L and M*). The mRNA levels, including *Ucp1*, *Ppargc1a*, *Cidea*, *Atp5b*, *Cox4*, *Cox8b*, *Cox7a*, *Mcad*, and *Fabp2*, were increased in the BAT from the Ang-(1-7)-treated *Lepr^db/db* mice (*Figure 4—figure supplement 1I*).

To sum up, our results suggested that the enhanced thermogenesis effect in the Ad-*Ace2* and Ang-(1-7) treated mice is caused by the increment of Ang-(1-7) levels, which demonstrates that Ang-(1-7) is crucial to the maintenance of thermogenesis.

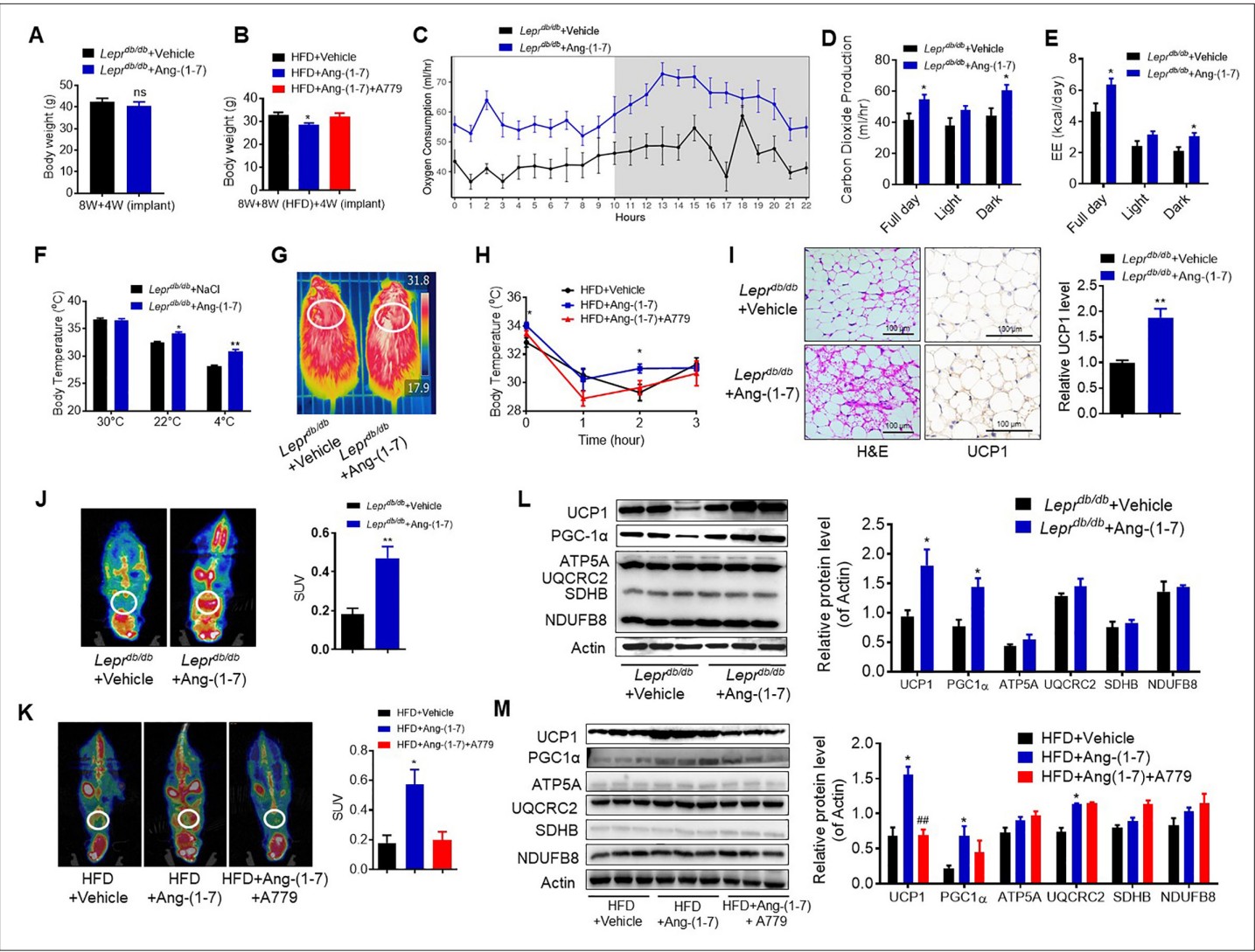

**Figure 4.** Ang-(1-7) promotes thermogenesis, BAT activity, and energy metabolism in the *Lepr^db/db* and the HFD-induced obese mice. Ang-(1-7) administration by subcutaneous implanted micro-osmotic pumps in the *Lepr^db/db* obese mice and the high-fat diet (HFD)-induced obese mice were used. The *Lepr^db/db* mice were treated with Ang-(1-7) by subcutaneous infusion of Ang-(1-7) or saline using osmotic mini-pumps for 4 weeks. Six-week-old male C57BL/6 J mice were used to develop obesity by HFD diet for 8 weeks, and the mice treated with Ang-(1-7), A779 (an Ang-(1-7) antagonist), or saline by osmotic mini-pumps at the 4th weeks post-HFD. (**A**) Body weight of *Lepr^db/db*+ Ang-(1-7) and *Lepr^db/db*+ Vehicle mice at the 4th week post micro-osmotic pumps implantation (n = 5–6/each group). (**B**) Body weight of HFD + Ang-(1-7), HFD + A779 and HFD + Vehicle mice at the 4th week post micro-osmotic pumps implantation (n = 4–7/each group). (**C–E**) Energy expenditure was evaluated by measurement of oxygen consumption (VO₂) (**C**), of carbon dioxide release (VCO₂) (**D**) and of energy expenditure (EE) (**E**) over a 24 hr period in *Lepr^db/db*+ Ang-(1-7) and *Lepr^db/db*+ Vehicle mice (n = 4–6/each group). (**F**) Core body temperature at 30 °C, 22°C and 4°C for 8 hr in *Lepr^db/db*+ Ang-(1-7) and *Lepr^db/db*-Vehicle mice (n = 5/each group). (**G**) Infrared thermal images at 22 °C in *Lepr^db/db*+ Ang-(1-7) and *Lepr^db/db*+ Vehicle mice. (**H**) Core body temperature at 4 °C for the indicated lengths of time in HFD + Ang-(1-7), HFD + A779 and HFD + Vehicle mice (n = 4–5/each group). (**I**) Representative H&E staining and UCP1 immunostaining from BAT sections of *Lepr^db/db*+ Ang-(1-7) and *Lepr^db/db*+ Vehicle mice exposure at 4 °C (n = 5/each group). (**J**) Representative Positron emission tomography–computed tomography (PET-CT) image and SUVs of *Lepr^db/db*+ Ang-(1-7) and *Lepr^db/db*+ Vehicle mice (n = 4/each group). (**K**) Representative PET-CT image and SUVs of HFD + Ang-(1-7), HFD + A779 and HFD + Vehicle mice (n = 3/each group). (**L**) Representative western blots showing the changes of key proteins of energy expenditure and thermogenesis in BAT of *Lepr^db/db*+ Ang-(1-7) and *Lepr^db/db*+ Vehicle mice exposure at 4 °C (n = 3/each group). (**M**) Representative western blots showing the changes of key proteins of energy expenditure and thermogenesis in BAT of HFD + Ang-(1-7), HFD + A779 and HFD + Vehicle mice exposure at 4 °C (n = 3/each group). Data represent mean ± SEM. *p < 0.05, **p < 0.01 *vs* Vehicle group by Student's t-test, ANCOVA, or one-way ANOVA.

The online version of this article includes the following source data and figure supplement(s) for figure 4:

**Source data 1.** Numerical quantification data for *Figure 4*.

**Figure supplement 1.** Ang-(1-7) promoted thermogenesis and energetic metabolism in BAT of *Lepr^db/db* mice during cold challenge.

**Figure supplement 1—source data 1.** Numerical quantification data for *Figure 4—figure supplement 1*.

## Ablation of *Mas1* impairs thermogenesis in brown adipose tissue

Since the Ang-(1-7), produced by ACE2, realized the function through the Mas1 receptor, these results above prompted us to hypothesize that the Mas1 receptor determines the effect of Ang-(1-7) in brown adipose tissue. Firstly, the HFD-induced *Mas1*$^{-/-}$ mice (low Ang-(1-7) action model) were used to assess the therapeutic effects (interventional effects) of Mas1 on energy metabolism. Although serum Ang-(1-7) levels were increased, the *Mas1*$^{-/-}$ mice had an impaired glucose tolerance, abnormal lipid profiles (*Figure 5A*, *Figure 5—figure supplement 1A-C*), and significantly increased body weight compared to the WT mice (*Figure 5B*). Meanwhile, the *Mas1*$^{-/-}$ mice exhibited decreased oxygen consumption (VO$_2$) (*Figure 5—figure supplement 1D*) without any changes in food and/or water intake as well as physical activity (*Figure 5—figure supplement 1E-G*). Moreover, the *Mas1*$^{-/-}$ mice had lower thermogenesis than the WT mice in either 22 °C or 4 °C (*Figure 5C, D and F*). PET-CT analysis illustrated that the *Mas1*$^{-/-}$ mice has less $^{18}$F-FDG uptake in BAT than the WT mice (*Figure 5E*). Consistently, the *Mas1*$^{-/-}$ mice displayed larger lipid droplets and reduced multilocular structures, and had reduced UCP1 expression compared with the WT mice (*Figure 5G*). Nevertheless, deletion of *Mas1* resulted in a striking repression of BAT thermogenic protein (UCP-1, UQCRC2, and SDHB) (*Figure 5H*) and genes (e.g. *Ucp1*, *Prmd16*, *Ppargc1a*, *Ppargc1b*, *Atp5b*, *Cox7a*, and *Cpt1a*) (*Figure 5I*).

To investigate the role of the Mas1 receptor in BAT, we generated BAT-specific *Mas1* knockout mice (*Mas1*$^{-/-}$ BAT transplanted mice). According to the previous studies (*Liu et al., 2013*; *Yuan et al., 2016*), firstly, BAT of the C57B/L6 recipient mice was removed from the interscapular region. Then, the BAT, which dissected from strain-, sex-, and age-matched *Mas1*$^{-/-}$ donor mice, was subcutaneously transplanted into the dorsal interscapular region of the C57B/L6 recipient mice (WT + *Mas1*$^{-/-}$-BAT). As controls, C57B/L6-recipient mice transplanted with C57B/L6 BAT (WT + WT BAT) and C57B/L6 epididymal white adipose tissue (eWAT) (WT + WT eWAT) were used as positive control and negative control, respectively. After the transplantation, the recipient mice were fed by HFD for 10 weeks.

Interestingly, compared with the WT + WT BAT control mice, the WT + *Mas1*$^{-/-}$-BAT mice showed greatly impaired HFD-induced insulin resistance. There is no significant difference between the WT + *Mas1*$^{-/-}$-BAT mice and the WT + WT eWAT mice in intraperitoneal glucose tolerance test (GTT) and the insulin tolerance test (ITT) (*Figure 5J and K*). Notably, *Mas1*$^{-/-}$ BAT transplantation also strikingly induced HFD-induced weight gain in the WT + *Mas1*$^{-/-}$-BAT mice compared with the WT + WT BAT control mice (*Figure 5L*).

More importantly, compared to the WT + WT BAT control mice, the WT + *Mas1*$^{-/-}$-BAT mice had decreased oxygen consumption (VO$_2$), carbon dioxide release (VCO$_2$), and energy expenditure (EE) (*Figure 5M–O*), along with normal food and/or water intake as well as physical activity (*Figure 5—figure supplement 1H-J*). Taken together, these results demonstrate that the Mas1 receptor can directly induce thermogenic program in brown adipose tissues.

## ACE2 pathway induces white fat browning and thermogenesis

Next, we investigated the impact of ACE2/Ang-(1-7) on the process of browning of WAT, a prominent feature in subcutaneous white adipose tissue (scWAT). Histological examination of scWAT from the Ad-*Ace2*-treated *Lepr*$^{db/db}$ obese mice showed a profound morphological transformation toward a BAT-like phenotype (smaller adipocytes with multiple lipid droplets) compared with the control (*Figure 6A*). Meanwhile, markers of brown adipocytes, such as *Ucp1*, were significantly increased in scWAT of the Ad-*Ace2*-treated *Lepr*$^{db/db}$ mice. A much greater induction of transcription factors, including *Prmd16*, *Ppargc1a* and *Ppargc1b*, occurred in the scWAT of the Ad-*Ace2*-treated group (*Figure 6B*). As expected, the Ang-(1-7) treated *Lepr*$^{db/db}$ mice had a similar browning effect as the Ad-*Ace2*-treated *Lepr*$^{db/db}$ mice in scWAT. Morphological brown-like adipocyte and thermogenic gene expression levels slight increasement were observed in the scWAT after Ang-(1-7) treatment (*Figure 6C and D*).

These alterations were restricted to scWAT, but not to epididymal white adipose tissue (eWAT). As shown in *Figure 6E*, the morphology and size of eWAT in the Ad-*Ace2*-treated *Lepr*$^{db/db}$ mice are same compared to the control (*Figure 6E*). Meanwhile, no significant increase in the thermogenic gene expression levels in the eWAT of the Ad-*Ace2*-treated *Lepr*$^{db/db}$ mice was found (*Figure 6F*).

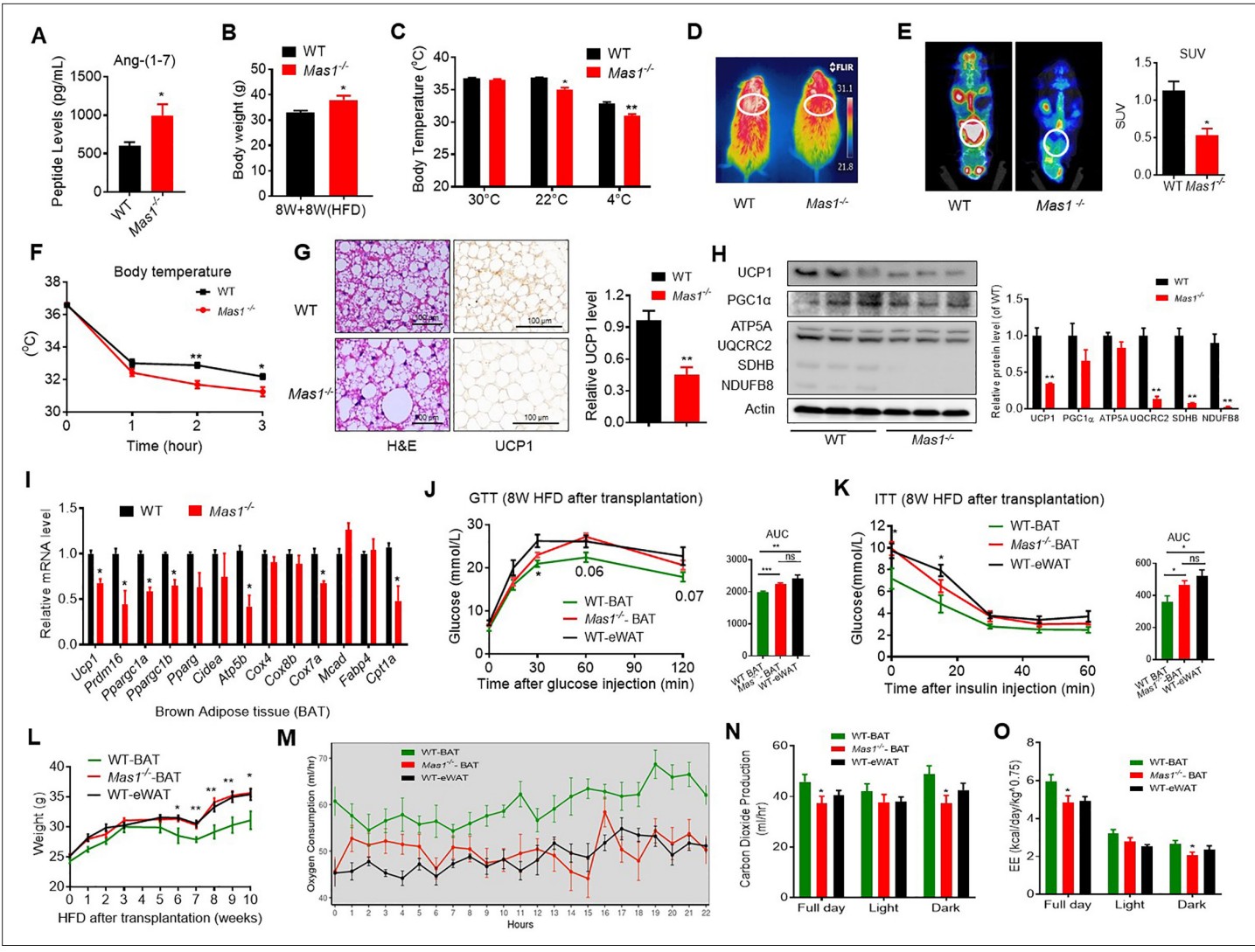

**Figure 5.** Ablation of *Mas1* impairs thermogenesis, BAT activity, and energetic metabolism. (**A–I**) Eight-week-old male *Mas1⁻ᐟ* mice and their WT (control) mice had a high-fat diet (HFD) for 8 weeks (*Mas1⁻ᐟ vs* WT). (**J–O**) BAT of C57B/L6 recipient mice was removed from the interscapular region. Then, the BAT dissected from *Mas1⁻ᐟ* donor mice, was subcutaneously transplanted into the dorsal interscapular region of C57B/L6 recipient mice (WT+ *Mas1⁻ᐟ*-BAT). C57B/L6 recipient mice transplanted with C57B/L6 BAT (WT+ WT BAT) and C57B/L6 epididymal white adipose tissue (eWAT) (WT+ WT eWAT) were used as control. The recipient mice were then fed an HFD immediately after the transplantation and continued for 10 weeks (WT+ *Mas1⁻ᐟ*-BAT *vs* WT+ WT BAT, WT+ WT eWAT). (**A**) Serum levels of Ang-(1-7) as determined by ELISA in *Mas1⁻ᐟ* and WT mice (n = 3–5/each group). (**B**) Body weight in *Mas1⁻ᐟ* and WT mice fed an HFD for 8 weeks (n = 4–5/each group). (**C**) Core body temperature at 30 °C, 22°C and 4°C for 8 hr in *Mas1⁻ᐟ* and WT mice (n = 5/each group).(**D**) Infrared thermal images at 22 °C in *Mas1⁻ᐟ* and WT mice. (**E**) Representative PET-CT image and SUVs of *Mas1⁻ᐟ* and WT mice (n = 3/each group). (**F**) Core body temperature at 4 °C for the indicated lengths of time in *Mas1⁻ᐟ* and WT mice (n = 4–5/each group). (**G**) Representative H&E staining and UCP1 immunostaining from BAT sections of *Mas1⁻ᐟ* and WT mice exposure at 4 °C (n = 5/each group). (**H**) Representative western blots showing the changes of key proteins in BAT of *Mas1⁻ᐟ* and WT mice exposure at 4 °C (n = 3/each group). (**I**) Relative mRNA levels of mitochondrial related genes, fatty acid oxidation related genes and transcription factors in BAT of in *Mas1⁻ᐟ* and WT mice exposure at 4 °C (n = 5–6/each group). (**J**) Intraperitoneal glucose tolerance test (GTT) and the average area under the curve (AUC) in WT+ *Mas1⁻ᐟ*-BAT, WT+ WT BAT and WT+ WT eWAT mice fed an HFD for 8 weeks after transplantation (n = 6–7/each group). (**K**) Insulin tolerance test (ITT) and AUC in WT+ *Mas1⁻ᐟ*-BAT, WT+ WT BAT and WT+ WT eWAT mice fed an HFD for 8 weeks after transplantation (n = 5–6/each group). (**L**) Body weight time course in WT+ *Mas1⁻ᐟ*-BAT, WT+ WT BAT and WT+ WT eWAT mice fed an HFD over 10 weeks after transplantation (n = 10/each group). (**M–O**) Energy expenditure was evaluated by measurement of oxygen consumption (VO₂) (**M**), of carbon dioxide release (VCO₂) (**N**) and of energy expenditure (EE) (**O**) over a 24 hr period in WT+ *Mas1⁻ᐟ*-BAT, WT+ WT BAT and WT+ WT eWAT mice (n = 5/each group). Data represent mean ± SEM. *p < 0.05, **p < 0.01 *vs* WT/WT-BAT group by Student's t-test, ANCOVA, or one-way ANOVA.

The online version of this article includes the following source data and figure supplement(s) for figure 5:

**Source data 1.** Numerical quantification data for *Figure 5*.

*Figure 5 continued on next page*

*Figure 5 continued*

**Figure supplement 1.** Ablation of *Mas1* impairs thermogenesis, BAT activity, and energetic metabolism.

**Figure supplement 1—source data 1.** Numerical quantification data for *Figure 5—figure supplement 1*.

## ACE2 pathway enhances thermogenesis via Akt and PKA signaling

We further investigated the molecular mechanisms through which ACE2 pathway regulates BAT. Firstly, we performed RNA-seq analysis on BAT isolated from the WT and the *Ace2* KO mice. Notable differences between the two are displayed as 3D-PCA analysis and heat map (*Figure 7—figure supplement 1A, B*). Consistent with the RT-PCR results of BAT in the *Ace2* KO mice, genetic deficiency of *Ace2* significantly altered expression of genes involved in fatty acid biosynthesis, lipid catabolism,

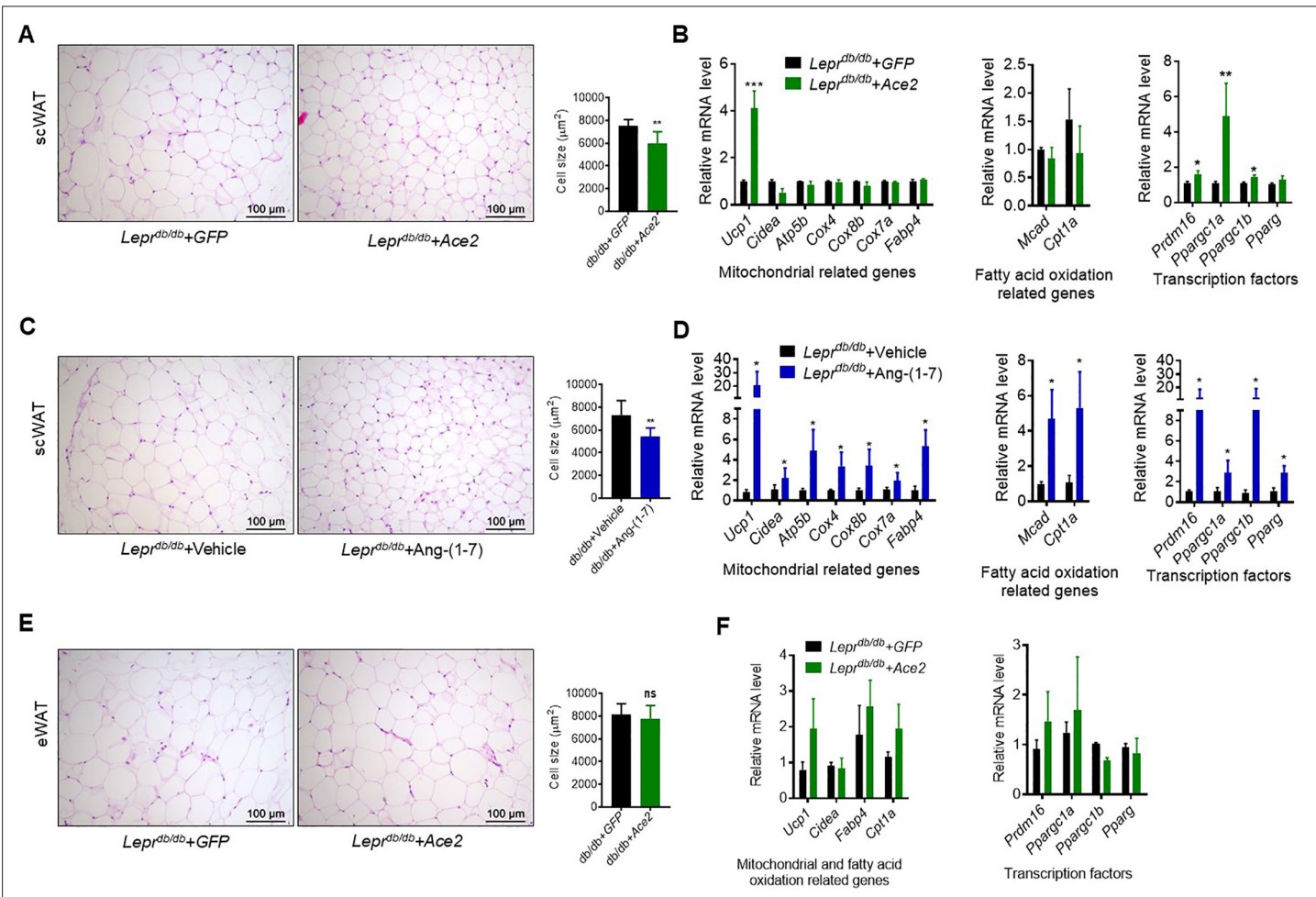

**Figure 6.** ACE2 pathway induced white adipose tissue browning in the *Lepr*^db/db^ obese mice. (**A**) Representative H&E staining from subcutaneous white adipose tissue (scWAT) sections of *Lepr*^db/db^+ Ace2 and *Lepr*^db/db^+ GFP mice exposure at 4 °C (n = 6–7/each group). (**B**) Relative mRNA levels of mitochondrial related genes, fatty acid oxidation related genes and transcription factors in scWAT of *Lepr*^db/db^+ Ace2 and *Lepr*^db/db^+ GFP mice exposure at 4 °C (n = 3–5/each group). (**C**) Representative H&E staining from scWAT sections of *Lepr*^db/db^+ Ang-(1-7) and Leprdb/db + Vehicle mice exposure at 4 °C (n = 6–7/each group). (**D**) Relative mRNA levels of mitochondrial related genes, fatty acid oxidation related genes and transcription factors in scWAT of *Lepr*^db/db^+ Ang-(1-7) and *Lepr*^db/db^+ Vehicle mice exposure at 4 °C (n = 4/each group). (**E**) Representative H&E staining from epididymal white adipose tissue (eWAT) sections of *Lepr*^db/db^+ Ace2 and *Lepr*^db/db^+ GFP mice exposure at 4 °C (n = 6–7/each group). (**F**) Relative mRNA levels of mitochondrial related genes, fatty acid oxidation related genes, and transcription factors in eWAT of *Lepr*^db/db^+ Ace2 and *Lepr*^db/db^+ GFP mice exposure at 4 °C (n = 4/each group). Data represent mean ± SEM. *p < 0.05, **p < 0.01 vs GFP/Vehicle group by Student's *t*-test.

The online version of this article includes the following source data for figure 6:

**Source data 1.** Numerical quantification data for *Figure 6*.

lipid biosynthesis, fatty acid beta-oxidation, and cholesterol biosynthesis in the BAT of the *Ace2* KO mice (*Figure 7—figure supplement 1C*).

Interestingly, we found that the expression level of Akt associated genes were significantly decreased in the *Ace2* KO mice compared with the WT mice (*Figure 7—figure supplement 1D*). The phosphorylation levels of Akt at residues Thr308 was significantly inhibited in BAT of the *Ace2* KO mice (*Figure 7A*). Furthermore, the phosphorylation levels of Akt were dramatically increased in BAT of the Ad-*Ace2*-treated mice (*Figure 7B*).

These results prompted us to consider whether ACE2 pathway regulates the function of BAT via Akt signaling. Thus, we treated primary brown adipocytes which isolated from mice with Ang-(1-7). We found phosphorylation of Akt was activated by Ang-(1-7), accompanied by UCP1 up-regulation and forkhead box-containing protein O subfamily-1 (FoxO1) phosphorylation (*Figure 7C*). MK2206, an Akt inhibitor (*Chorner and Moorehead, 2018*; *Matsuzaki et al., 2018*), suppressed Ang-(1-7)-induced UCP1 up-regulation and FoxO1 phosphorylation (*Figure 7C*). Compared to Ang-(1-7)-treated primary brown adipocytes cells, MK2206 down-regulated the mRNA levels of *Ucp1*, *Ppargc1a*, *Cidea*, *Atp5b*, *Fabp4*, and *Cpt1a* genes (*Figure 7D*). More importantly, Ang-(1-7)-treated primary brown adipocytes exhibited higher OCR, and MK2206 inhibited OCR in the Ang-(1-7) treated primary brown adipocytes (*Figure 7E*). Accordingly, the *Ace2*-overexpressing primary brown adipocytes cells showed a similar result. MK2206 suppressed ACE2 induced up-regulation of protein (UCP1 and phosphorylated FoxO1) (*Figure 7—figure supplement 2A*) and mRNA (*Ucp1* and *Cpt1a*) (*Figure 7—figure supplement 2B*), as well as action on OCR (*Figure 7—figure supplement 2C*). These results suggest that the Akt signaling are required for the thermogenic activity of ACE2 pathway.

After the determination that ACE2 pathway regulates adaptive thermogenesis through the Akt signaling, we paid attention on whether this program could still be provoked by protein kinase A (PKA) signaling, a pathway known to be involved in the canonical thermogenic activation of fat cells. Interestingly, we found that the phosphorylation level of PKA was significantly inhibited in BAT of the *Ace2* KO mice (*Figure 7F*). Similar results appeared in the *Mas1* KO mice (*Figure 7—figure supplement 2D*). However, the phosphorylation level of PKA was increased in BAT of the Ad-*Ace2*-treated *Lepr^{db/db}* mice (*Figure 7G*).

To further elucidate the study on the mechanism of Ang-(1-7)-induced PKA signaling, we administrated Ang-(1-7)-treated primary brown adipocytes with PKA and adenylylcyclase inhibitors, simultaneously. Firstly, the Ang-(1-7)-induced PKA signaling was validated in primary brown adipocytes (*Figure 7—figure supplement 2E*). However, H89, a PKA inhibitor, significantly blunted the Ang-(1-7)-induced mRNA levels (*Ucp1*, *Cidea*, and *Fabp4*) (*Figure 7H*) and protein levels (UCP1 and PGC1α) (*Figure 7I*). Similar effects were observed by using SQ-22536, an adenylylcyclase inhibitor on the formation of intracellular cAMP (*Figure 7H,I*). As expected, the Ang-(1-7)-treated primary brown adipocytes exhibited higher OCR, and H89 inhibited OCR in the Ang-(1-7)-treated primary brown adipocytes (*Figure 7J*). H89 also suppressed ACE2-induced maximal respiration (*Figure 7—figure supplement 2F*). Our results thus strongly suggest that the PKA signaling is important for the thermogenic activity of ACE2 pathway.

Combine with the above result, these data indicated that Ang-(1-7) treatment induced respiration at least partly through Akt and PKA signaling pathway. Notably, treatment with PKA inhibitors on non-Ang-(1-7) stimulated cells results in very little change in OCR, whereas treatment on Ang-(1-7) stimulated cells results in reversion to normal OCR levels (*Figure 7J*). In contrast, treatment of non-Ang-(1-7) stimulated cells with AKT inhibitors led to an equivalent level of decrease in OCR as in Ang-(1-7)-treated cells (*Figure 7E*), suggesting that AKT inhibition may generically decrease OCR in adipocytes, whereas inhibiting PKA only affects the OCR of Ang-(1-7)-treated cells. PKA is more likely responsible in the changes in mitochondrial activity due to Ang-(1-7) stimulation than AKT signaling. More importantly, our data showed that co-treatment with Akt and PKA inhibitors have an additive effect in *Ace2*-overexpressing primary brown adipocytes cells (*Figure 7—figure supplement 2G*).

Furthermore, in Ang-(1-7)-treated primary brown adipocytes, there is no significant difference between A779 and PKA/Akt inhibitor treatment on OCR (*Figure 7—figure supplement 2H*). This data indicated that A779 treatment plays similar effect to PKA/Akt inhibitor on mitochondrial respiration in Ang-(1-7) treated cells.

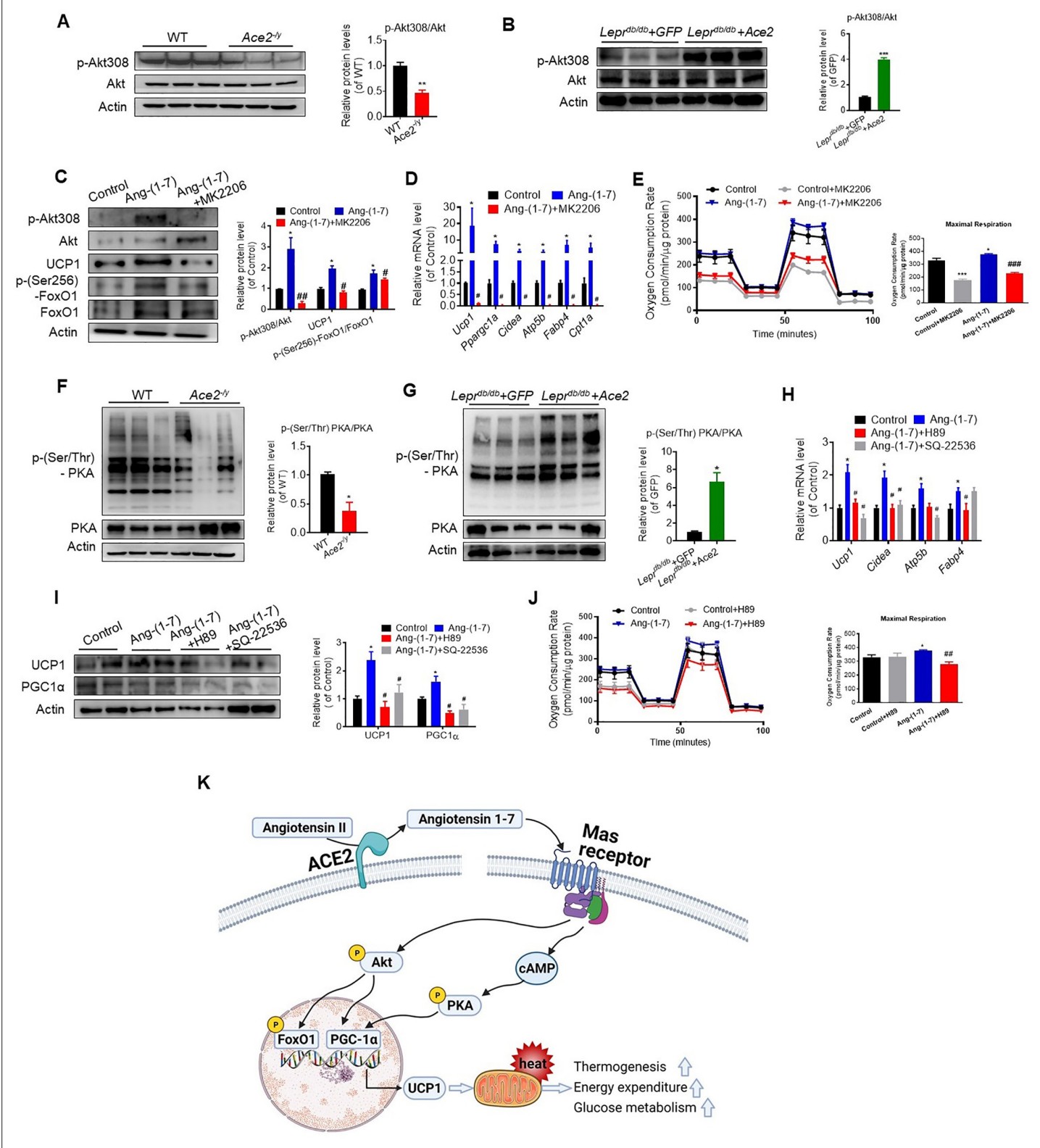

**Figure 7.** ACE2 pathway induces a thermogenesis program through the Akt signaling and the PKA signaling. Primary brown adipocytes were isolated, cultured, and treated with Ang-(1-7) ($10^{-6}$ M) for 24 hr, Akt inhibitor MK2206 (20 µM) for 24 hr, PKA inhibitor H89 (30 µM) for 24 hr, or adenylylcyclase inhibitor SQ-22536 (10 µM) for 24 hours. (**A, B**) Representative western blots showing the changes of p-Akt308 and Akt in BAT of $Ace2^{-/y}$ (**A**) and $Lepr^{db/db}$+ Ace2 mice (**B**) exposure at 4 °C (n = 3/each group). (**C**) Representative western blots showing the Akt, UCP1 and forkhead box protein O 1 (FoxO1)

*Figure 7 continued on next page*

*Figure 7 continued*

Changes (n = 3/each group). (**D**) Relative mRNA levels of thermogenic and mitochondrial genes (n = 4–6/each group). (**E**) Continuous measurement of oxygen consumption rate (OCR) in Ang-(1-7) and MK2206 treated primary brown adipocytes. Oxygen consumption was performed under basal conditions, following the addition of oligomycin (1 µM), the pharmacological uncoupler FCCP (1 µM) or the Complex III and I inhibitor antimycin A and rotenone (0.5 µM) (n = 3–4/each group). (**F, G**) Representative western blots showing the p-PKA and PKA changes in BAT of *Ace2^-/y* (**F**) and *Lepr^db/db* + *Ace2* mice (**G**) exposure at 4 °C (n = 3/each group). (**H**) Relative mRNA levels of thermogenic and mitochondrial genes (n = 4–6/each group). (**I**) Representative western blots showing the UCP1 and PGC1α changes (n = 3/each group). (**J**) Continuous measurement of OCR in Ang-(1-7) and H89-treated primary brown Adipocytes (n = 3–5/each group). Data represent mean ± SEM. *p< 0.05, **p < 0.01 *vs* GFP/WT group by Student's *t*-test. *p < 0.05, **p < 0.01 vs control group, # p < 0.05, ## p < 0.01 vs Ang-(1-7) group by one-way ANOVA. (**K**) Mechanisms involved in ACE2 pathway activation-induced improvement of BAT function.

The online version of this article includes the following source data and figure supplement(s) for figure 7:

**Source data 1.** Numerical quantification data for *Figure 7*.

**Figure supplement 1.** RNA-Seq analysis of primary brown adipocytes from *Ace2^-/y* and WT mice.

**Figure supplement 1—source data 1.** Numerical quantification data for *Figure 7—figure supplement 1*.

**Figure supplement 2.** Ang-(1-7)/ACE2 regulate thermogenesis through Akt and PKA signaling in BAT.

**Figure supplement 2—source data 1.** Numerical quantification data for *Figure 7—figure supplement 2*.

## Discussion

RAS is classically known to regulate blood pressure and maintain water and electrolyte balance. It also plays as a crucial role in metabolic disorders, such as obesity and insulin resistance (*Das, 2016*). Comprehensive understanding of the complexly biological function of the RAS remains a major biomedical challenge. Recently, a research group separately reported that ACE2 and Ang-(1-7) exert anti-obesity effect by BAT in HFD-induced obesity mice (*Kawabe et al., 2019*; *Morimoto et al., 2018*). In ACE2 pathway, ACE2, Ang-(1-7), and Mas1 receptor interact and antagonize each other's actions. As these two studies only used gain-of-function approaches, they cannot clarify the specific physiological role of ACE2-Ang1-7-Mas1 axis in the thermogenesis brown adipose tissue. Therefore, it is critically important and urgent to systematically analyze the function of these three elements using reverse genetics approaches.

In our study, we used seven mice models, *Ace2* KO mice, Ad-*Ace2 Lepr^db/db* and HFD mice, Ang-(1-7) treated *Lepr^db/db* and HFD mice, *Mas1* KO mice and BAT-specific *Mas1* knockout mice (*Mas1^-/-* BAT transplanted mice), respectively. Based on a series of functional assays in these mouse models and primary brown adipocytes, we effectively confirmed that the ACE2 pathway regulates glucose and lipid homeostasis. Furthermore, ACE2 pathway maintains thermogenesis and systemic energy metabolism. Molecular analyses, including the use of several inhibitors of Akt and PKA, demonstrate that the effects of ACE2 pathway on brown adipocytes are mediated by both the Akt signaling and the PKA signaling, resulting in the activation of PGC1α, followed by activation of UCP1 (*Figure 7K*). These findings significantly expand our understanding of the biological function of RAS. Furthermore, our results propose a new concept that the ACE2 pathway can improve obesity and the associated metabolic disorders.

BAT, which utilizes glucose and fatty acids for thermogenesis, contains large number of mitochondria and promotes thermogenesis by mitochondrial respiration through UCP1. BAT-specific UCP1, localized in the inner mitochondrial membrane, plays a fundamental role in thermogenesis. In response to stimulation, activation of PGC1α up-regulates the expression of BAT-specific UCP1, which dissipates the proton motive force across the inner mitochondrial membranes, and consequentially producing ATP (*Sambeat et al., 2016*; *Schreiber et al., 2017*). On the other hand, PGC1α induces the acquisition of BAT features, including the expression of mitochondria and fatty acid-oxidation and thermogenic genes (*Puigserver et al., 1998*; *Tiraby et al., 2003*). We found that the mRNA levels of *Ucp1*, *Ppargc1a*, mitochondrial program and fatty acid oxidation related genes (*Ppargc1b*, *Atp5b*, *Cox7a*, *Cox8b*, *Fabp4*, and *Cpt1a*) were up-regulated in the *Ace2* overexpression and the Ang-(1-7)-treated *Lepr^db/db* mice, whereas down-regulated in the *Ace2* KO or the *Mas1* KO mice. These results supported that PGC1α and *UCP1* might be critical for the effects of ACE2 pathway on thermogenesis.

We also investigated the underlying mechanisms of ACE2 pathway on the regulation of BAT via the Akt signaling and the PKA signaling (*Figure 7K*).

First, we verified the Akt signaling in the downstream of ACE2 pathway. Akt has a critical function in cell survival and energy balance. Multiple pieces of evidence show that activation of PI3K is followed by the activation of Akt, which in turn triggers a complex cascade of events that include the inhibition of FoxO1 transcription factors and thus the activation of *Ucp1* and its transcriptional regulator PGC1α (*Nakae et al., 2008*; *Ortega-Molina et al., 2012*). In human and 3T3-L1 preadipocytes, Ang-(1-7)-Mas1 signaling promotes adipogenesis via activation of PI3K-Akt signaling (*Than et al., 2013*). AT2R activation induces white adipocyte browning by increasing PPARγ expression, at least in part, via PI3K-Akt signaling pathways (*Than et al., 2017*). We previously reported that ACE2 and Ang-(1-7) can activate Akt signaling to ameliorate hepatic steatosis (*Cao et al., 2016*; *Cao et al., 2014*). In the present study, the *Ace2* KO and the *Mas1* KO mice displayed a strong decrease in Akt S308 phosphorylation in BAT. The *Ace2* over-expression or the Ang-(1-7) treatment activated Akt S308 phosphorylation in BAT. Furthermore, the effect of ACE2-Ang-(1-7) on primary brown adipocytes can be attenuated by Akt inhibitor. These results suggest that the Akt signaling might also play a role in ACE2 pathway related regulation of BAT function.

Second, we verified the PKA signaling in the downstream of ACE2 pathway. The Mas1 receptor was shown to constitutively couple to Gαs, including Gαi, Gαq, and Gα12/13proteins (*Dias-Peixoto et al., 2008*; *Gomes et al., 2012*; *Tirupula et al., 2014*). In the kidney, Ang-(1-7) treatment increased cAMP levels and activated PKA through Gαs activation by the Mas1 receptor (*Liu et al., 2012b*; *Magaldi et al., 2003*). Ang-(1-7) regulates insulin secretion through a Mas1-dependent cAMP signaling pathway (*Sahr et al., 2016*). It is well known that norepinephrine released from the sympathetic nerves is a powerful stimulator of BAT. Norepinephrine activates BAT thermogenic program via PKA signaling, followed by the UCP1-mediated proton uncoupling (*Su et al., 2017*). In this study, the PKA signaling in BAT was changed significantly by ACE2 pathway in mice model. In addition, the effect of ACE2 pathway on primary brown adipocytes can be depressed by cAMP and PKA inhibitor.

Previously, we demonstrate that the ACE2 pathway is involved in the regulation of glucose and lipid homeostasis with limited understanding of the underlying mechanisms (*Cao et al., 2014*; *Liu et al., 2012a*; *Niu et al., 2008*; *Shi et al., 2018*; *Zhang et al., 2016*). Here, for the first time, we provide evidence that the alteration in glucose and lipid homeostasis is associated with the change in maintaining brown adipocyte function for the facilitation of energy expenditure. In summary, the ACE2 pathway regulates BAT function and systemic energy metabolisms which is a potential treatment target for metabolic disorders including metabolic syndrome, diabetes, dyslipidemia, and fatty liver.

# Materials and methods

**Key resources table**

| Reagent type (species) or resource | Designation | Source or reference | Identifiers | Additional information |
|---|---|---|---|---|
| Genetic reagent (*Mus musculus*) | WT C57BL/6 J | GemPharmatech. Co., Ltd | JAX 000664 RRID: IMSR_JAX:000664 | |
| Genetic reagent (*Mus musculus*) | BKS-db (*Lepr^db/db^*) | GemPharmatech. Co., Ltd | | |
| Genetic reagent (*Mus musculus*) | *Ace2* KO | Institute of Laboratory Animal Science, Chinese Academy of Medical Sciences | | |
| Genetic reagent (*Mus musculus*) | *Mas1* KO | GemPharmatech. Co., Ltd | | |
| Transfected construct (*Mus musculus*) | Ad-*Ace2*-eGFP | SinoGenoMax | | |
| Chemical compound, drug | Ang-(1-7) | MCE | 51833-78-4 | |
| Chemical compound, drug | A779 | Selleck | 159432-28-7 | |

*Continued on next page*

*Continued*

| Reagent type (species) or resource | Designation | Source or reference | Identifiers | Additional information |
|---|---|---|---|---|
| Chemical compound, drug | FCCP | Sigma-Aldrich | C2920 | |
| Chemical compound, drug | Oligomycin A | Sigma-Aldrich | 75351–5 MG | |
| Chemical compound, drug | Rotenone | Sigma-Aldrich | R8875-1G | |
| Chemical compound, drug | MK2206 | Selleck | 1032350-13-2 | |
| Chemical compound, drug | H89 | Selleck | 130964-39-5 | |
| Chemical compound, drug | SQ-22536 | Selleck | 17318-31-9 | |
| Other | Chow, 60% HFD | Research Diets | D12492 | |
| Antibody | Anti-UCP1 (rabbit polyclonal) | Abcam | #10983 RRID: AB_2241462 | (1:1000) |
| Antibody | Anti-PGC1α (rabbit polyclonal) | Abcam | #54,481 RRID: AB_881987 | (1:1000) |
| Antibody | Anti-OXPHOS | Abcam | #110413 RRID: AB_2629281 | (1:1000) |
| Antibody | Anti-Mas1 (rabbit polyclonal) | Alomone | #AAR-013 RRID: AB_2039972 | (1:1000) |
| Antibody | Anti-Akt (rabbit polyclonal) | Cell signaling | #9272 RRID: AB_329827 | (1:1000) |
| Antibody | Anti-p-Akt308 (rabbit monoclonal) | Cell signaling | #13038 RRID: AB_2629447 | (1:1000) |
| Antibody | Anti-PKA (rabbit polyclonal) | Cell signaling | #4782 RRID: AB_2170170 | (1:1000) |
| Antibody | Anti-p-PKA (rabbit polyclonal) | Cell signaling | #9,621 RRID: AB_330304 | (1:1000) |
| Antibody | Anti-ACE2 (rabbit monoclonal) | Cell signaling | #92,485 | (1:1000) |
| Antibody | Actin (rabbit monoclonal) | Cell signaling | #4,970 RRID: AB_2223172 | (1:1000) |
| Sequence-based reagent | *Cidea*_F | Invitrogen | RT-qPCR primer | TCCTATGCTGCACAGATGACG |
| Sequence-based reagent | *Cidea*_R | This paper | RT-qPCR primer | TGCTCTTCTGTATCGCCCAGT |
| Sequence-based reagent | *Ppargc1*a_F | This paper | RT-qPCR primer | GCACCAGAAAACAGCTCCAAG |
| Sequence-based reagent | *Ppargc1*a_R | This paper | RT-qPCR primer | CGTCAAACACAGCTTGACAGC |
| Sequence-based reagent | *Ucp1*_F | This paper | RT-qPCR primer | TCTCAGCCGGCTTAATGACTG |
| Sequence-based reagent | *Ucp1*_R | This paper | RT-qPCR primer | GGCTTGCATTCTGACCTTCAC |
| Sequence-based reagent | *Prdm16*_F | This paper | RT-qPCR primer | ACACGCCAGTTCTCCAACCTGT |

*Continued on next page*

*Continued*

| Reagent type (species) or resource | Designation | Source or reference | Identifiers | Additional information |
|---|---|---|---|---|
| Sequence-based reagent | *Prdm16_R* | This paper | RT-qPCR primer | TGCTTGTTGAGGGAGGAGGTA |
| Software, algorithm | GraphPad Prism Software | GraphPad Software, La Jolla, CA, USA | Version 8.0.0 for Windows RRID: SCR_002798 | |
| Software, algorithm | ANCOVA | PMID:30017358 | https://calrapp.org/ | |

## Mice

Obese BKS-db (*Lepr^{db/db}*) male mice, wild-type mice and *Mas1* KO mice were purchased from Nanjing Biological Medicine Research Institute, Nanjing University, China. Male C57BL/6 J mice were purchased from Vital River Laboratory Animal Technology (Beijing, China). *Ace2* KO mice have been previously described (*Niu et al., 2008*).

The obese diabetic *Lepr^{db/db}* mice at 7–8 weeks of age were used. Adenovirus ($5 \times 10^8$ particle forming units (pfu) in a total volume of 100 µL of 0.9% wt/vol saline) was introduced into the *Lepr^{db/db}* mice by tail vein injection. The ad-*Ace2 Lepr^{db/db}* mice were used at the 6th day post-virus injection. The *Lepr^{db/db}* mice were treated with Ang-(1-7) by subcutaneous infusion of Ang-(1-7) (100 ng/kg/min) or saline using osmotic mini-pumps (Alzet-Durect, Cupertino, CA, USA Model #1004) for 4 weeks.

Six-week-old male C57BL/6 J mice were used to develop obesity by high-fat (HFD) diet (60 kcal% fat) (Research Diets, New Brunswick, NJ, USA) for 8 weeks, and the mice treated with Ang-(1-7) by osmotic mini-pumps at the 5th weeks post-HFD. Eight- to 10 weeks old male *Ace2* KO mice and WT controls, *Mas1* KO and WT controls were fed HFD diet for 8 weeks before experimental analysis. The mice were housed in a room at controlled temperature (23°C ± 1°C) with a 12-hr light-dark cycle. All animals were handled in accordance with the protocol approved by the Ethics Committee of Animal Research at Beijing Tongren Hospital, Capital Medical University, Beijing, China.

## BAT transplantation

According to the methods described previously (*Liu et al., 2013*; *Yuan et al., 2016*), BAT was removed from the interscapular region of 8 week old *Mas1* KO mice or C57BL/6 mice donor mouse and implanted into the interscapular region of recipient mice. BAT of C57B/L6 recipient mice was removed from the interscapular region. After cervical dislocation of donor mice, the BAT or eWAT (also from the epididymal fat pad of 8-week-old C57BL/6 mice) was removed and peripheral white fat was excluded, and then the remaining BAT (0.2 g) or eWAT (0.2 g) was washed with sterile PBS and transplanted into the interscapular region of recipients as quickly as possible. Recipient mice were anesthetized by ip injection with 400 mg/kg body weight avertin, and then BAT or eWAT was transplanted underneath the skin. The recipient mice were then fed an HFD, which began immediately after the transplantation and continued for 10 weeks.

## Adipocyte oxygen consumption rate (OCR) measurement

Primary brown adipose cells were isolated and cultured for 3 days before plated in XF cell culture microplates (Seahorse Bioscience). Cells (10,000 cells) were seeded in each well and each sample has eight replicates. After 6 days of differentiation, cultured adipocytes were washed twice and pre-incubated in XF medium for 2 hr at room temperature. The oxygen consumption rate was measured by the XF extracellular flux analyzer (Seahorse Biosciences). The results were normalized to protein content in each well. The process was cycled three times for baseline and drug injection measurement. Each cycle consisted of 3 min of mix, 2 min of delay, and 3 min of measurement time. The concentrations of the injection compounds used were Oligomycin (1 µM), FCCP (1 µM), and Rotenone/antimycin A (0.5 µM).

## Glucose tolerance test (GTT)

Mice were fasted for 16 hours (17:00–9:00) with free access to drinking water. Glucose (1.0 g/kg for the *Lepr^{db/db}* mice and 2.0 g/kg for the HFD mice) was administered intraperitoneally (i.p.), and blood

glucose levels were measured immediately 0, 15, 30, 60, and 120 min after glucose injection by using an Accu-Chek glucose monitor (Roche Diagnostics Corp).

## Insulin tolerance test (ITT)

ITT was performed by injecting intraperitoneally 0.75 IU/kg of insulin at mice fasted for 1 hr and measured blood glucose levels at 0, 15, 30, 60, 90, and 120 min post injection by using an Accu-Chek glucose monitor (Roche Diagnostics Corp).

## RNA extraction and quantitative real-time RT-PCR

Total RNA was isolated using TRIzol reagent (Invitrogen, Carlsbad, CA, USA) according to the manufacturer's instructions. A total of 500 ng of RNA was used as the template for the first-strand cDNA synthesis using ReverTra Ace qPCR RT Kit (TOYOBO, Osaka, Japan) in accordance with the manufacturer's protocol. The transcripts were quantified using Light Cycler 480 Real-Time PCR system (Roche, Basel, Switzerland). Primers were designed using Primer Quest (Integrated DNA Technologies, Inc).

## Positron emission tomography–computed tomography (PET-CT)

Siemens Inveon Dedicated PET (dPET) System and Inveon Multimodality (MM) System (CT/SPECT) (Siemens Preclinical Solutions) was used to detect PET-CT imaging at Chinese Academy of Medical Sciences. According to the previously studies (*Liu et al., 2013*; *Yuan et al., 2016*), mice were allowed to fast overnight and were lightly anesthetized with isoflurane, and then followed by a tail vein injection of 18F-FDG (500 mCi). Sixty minutes after the injection of the radiotracer, the mice were subjected to PET/CT analysis. A 10 min CT X-ray for attenuation correction was scanned before PET-CT scan. Static PET-CT scans were acquired for 10 min, and images were reconstructed by an OSEM3D algorithm followed by Maximization/Maximum a Posteriori (MAP) or Fast MAP provided by Inveon Acquisition Workplace (IAW) software. The 3D regions of interest (ROIs) were drawn over the guided CT images, and the tracer uptake was measured using Inveon Research Workplace (IRW) (Siemens) software. Individual quantification of the 18F-FDG uptake in each of the ROI was calculated. The data for the accumulation of 18F-FDG on micro-PET images were expressed as the standard uptake values (SUVs), which were determined by dividing the relevant ROI concentration by the ratio of the injected activity to the bodyweight.

## Western blot analysis

Tissues were dissolved in RIPA buffer (150 mM sodium chloride, 1.0% Triton X-100, 0.5% sodium deoxycholate, 0.1% SDS, 50 mM Tris, protease and phosphatase inhibitor mixture (Roche Diagnostics)). Protein concentrations were determined using a BCA assay kit (Pierce Diagnostics). Protein was separated by 10% (wt/vol) SDS/PAGE, transferred to a PVDF membrane (Millipore), blocked in 5% (wt/vol) skim milk in TBST (0.02 M Trisbase, 0.14 M Vehicle, 0.1% Tween 20, pH 7.4), and incubated with primary antibodies overnight at 4 °C and then incubated with secondary antibodies conjugated with HRP. The following primary antibodies were used: anti-UCP1 (ab10983, Abcam), anti-PGC1α (ab54481, Abcam), anti-OXPHOS (ab110413, Abcam), anti-Mas1 (AAR-013, Alomone labs), anti-Akt (#9272, cell signaling technology), anti-p-Akt308 (#13038, cell signaling technology), anti-FoxO1 (#2880, cell signaling technology), anti-p-FoxO1 (#84192, cell signaling technology), anti-PKA (#4782, cell signaling technology), anti-p-PKA (#9621, cell signaling technology), anti-ACE2 (#92485, cell signaling technology), and actin (#4970, Cell Signaling Technology). Signals were detected with Super Signal West Pico Chemiluminescent Substrate (Pierce).

## Histology and immunofluorescence analysis

Tissues fixed in 4% paraformaldehyde were sectioned after being paraffin embedded. Multiple sections were prepared and stained with hematoxylin and eosin for general morphological observations. Cells grown on poly-L-lysine (Sigma)-pretreated coverslips were fixed with 4% paraformaldehyde. Immunofluorescence staining was performed according to the standard protocol using the following antibodies and dilutions: UCP1 (1:100 dilution; Santa Cruz Biotechnologies), MitoTracker Red (1:1,000 dilution; Invitrogen). Incubations were performed overnight in a humidified chamber at 4 °C. 40, 6-diamidino-2-phenylindole staining was used to mark the cell nuclei. The images were acquired by microscope (DS-RI1; Nikon).

## Metabolic rate and physical activity

Oxygen consumption and physical activity were determined with a TSE LabMaster, as previously described (*Chi and Wang, 2011*). Mice were acclimated to the system for 24 hr, and then $VO_2$ and $VCO_2$ were measured during the next 24 hr. Voluntary activity of each mouse was measured with an optical beam technique (Opto-M3, Columbus Instruments, Columbus, OH, USA) over 24 hr and expressed as 24 hr average activity. Energy expenditure and respiratory exchange ratio were then calculated (*Liu et al., 2013*).

## RNA-Seq analysis

Total RNA was extracted from *Ace2* KO or WT primary brown adipocytes by Trizol reagent (Invitrogen), respectively. Extracted RNA samples were sent to Novel Bioinformatics company (Shanghai, China) for RNA-seq. RNA with RIN >8.0 is right for cDNA library construction. The cDNA libraries were processed for the proton sequencing according to the commercially available protocols. Data were submitted to the GEO archive. Fisher's exact test was calculated to select the significant pathway, and the threshold of significance was defined by p-value and false discovery rate (FDR) (*Dupuy et al., 2007*).

## Infrared thermography and core temperature

Mice were exposed to a cold chamber (4 °C) with one mouse per cage for up to 6 hr, with free access to food and water. An infrared digital thermographic camera was used to taken images (E60: Compact Infrared Thermal Imaging Camera; FLIR). The images were analyzed by FLIR Quick Report software (FLIR ResearchIR Max 3.4; FLIR). A rectal probe connected to a digital thermometer was used to measure core body temperature (Yellow Spring Instruments).

## Statistical analysis

All of the data are presented as the mean ± SEM. The data were analyzed by Student's *t*-test or one-way ANOVA (with Bonferroni post-hoc tests to compare replicate means) when appropriate. Statistical comparisons were performed using Prism5 (GraphPad Software, San Diego, CA). The CLAMS data were performed using ANCOVA on webtool (https://calrapp.org/) (*Mina et al., 2018*). p Values less than 0.05 were considered statistically significant. Representative results from at least three independent experiments are shown unless otherwise stated.

## Acknowledgements

This work was supported by grants from National Key R&D Program of China (2017YFC0909600) and National Natural Science Foundation of China (81561128015, 81471014) to Jinkui Yang, National Natural Science Foundation of China (81670774, 82070850) and Beijing Natural Science Foundation (7162047) to Xi Cao.

## Additional information

### Funding

| Funder | Grant reference number | Author |
| --- | --- | --- |
| National Natural Science Foundation of China | 81930019 | Jin-Kui Yang |
| National Natural Science Foundation of China | 81561128015 | Jin-Kui Yang |
| National Natural Science Foundation of China | 81670774 | Xi Cao |
| National Natural Science Foundation of China | 82070850 | Xi Cao |

| Funder | Grant reference number | Author |
|---|---|---|
| Beijing Natural Science Foundation | 7162047 | Xi Cao |

The funders had no role in study design, data collection and interpretation, or the decision to submit the work for publication.

## Author contributions

Xi Cao, Conceptualization, Formal analysis, Funding acquisition, Investigation, Project administration, Writing – original draft, Writing – review and editing; Ting-Ting Shi, Conceptualization, Formal analysis, Investigation, Writing – original draft; Chuan-Hai Zhang, Conceptualization, Investigation, Methodology, Writing – original draft, Writing – review and editing; Wan-Zhu Jin, Aimin Xu, Conceptualization; Li-Ni Song, Yi-Chen Zhang, Jing-Yi Liu, Fang-Yuan Yang, Investigation; Charles N Rotimi, Formal analysis, Writing – review and editing; Jin-Kui Yang, Conceptualization, Funding acquisition, Project administration, Writing – review and editing

## Author ORCIDs

Xi Cao http://orcid.org/0000-0001-5447-6766
Chuan-Hai Zhang http://orcid.org/0000-0001-7644-2436
Jin-Kui Yang http://orcid.org/0000-0002-5430-2149

## Ethics

All animal protocols used in this study were reviewed and approved by the Ethics Committee of Animal Research at Beijing Tongren Hospital, Capital Medical University (#2017-0107).

## Decision letter and Author response

Decision letter https://doi.org/10.7554/eLife.72266.sa1
Author response https://doi.org/10.7554/eLife.72266.sa2

---

# Additional files

## Supplementary files

- Transparent reporting form
- Source data 1. Original files for gels or blots.
- Source data 2. Powerpoint files containing gels or blots.

## Data availability

All data generated or analysed during this study are included in the manuscript and supporting file.

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
