## [Editor Report]

Authors have found that ACE2 is highly expressed in brown adipose tissue (BAT), indicating ACE2 pathway as a critical regulator in the maintenance of thermogenesis and energy expenditure. Identifying new regulators of energy homeostasis authors have shed light on novel potential therapeutic targets for the treatment of metabolic disorders.

---

## [Decision Letter]

**Decision letter after peer review:**

Thank you for submitting your article "ACE2 Pathway Regulates Thermogenesis and Energy Metabolism" for consideration by *eLife*. Your article has been reviewed by 2 peer reviewers, and the evaluation has been overseen by a Reviewing Editor and Carlos Isales as the Senior Editor. The reviewers have opted to remain anonymous.

Essential revisions:

1) Body temperature experiments in ACE2-overexpressing mice in the HFD model

2) Experiments to clarify the relationship between PKA and AKT signaling, RAS signaling and respiration

*Reviewer #1 (Recommendations for the authors):*

1) The authors provide ample evidence that both AKT signaling and PKA signaling contribute to RAS signaling upregulation of UCP1 and mitochondrial respiration. In their model, they have the AKT and PKA pathways as unrelated inputs into mitochondrial respiration. However, primary BATs treated with PKA and AKT inhibitors seem have a similar total loss of RAS-induced oxygen consumption. This either implies that they are either part of the same pathway (not unrelated) or that AKT and PKA inhibitors are shutting off respiration in general and that this is not related to RAS-induced respiration in particular. Some experiments to clarify the relationship between PKA and AKT signaling, RAS signaling and respiration would be helpful. For example, does co-treatment with AKT and PKA inhibitors have an additive effect? Do these inhibitors have an effect on respiration independent of Ang-(1-7) treatment. It may be interesting to see if Ang-(1-7) treated cells respond differently to A779 treatment than PKA or AKT inhibition in terms of mitochondrial respiration to determine the magnitude of the effect on respiration of PKAi/AKTi are similar to the effect of a RASi. It may be valuable to use adipocytes from the diabetic mice and induce Ace2 in those mice, then observe how AKT and PKA inhibitors effect respiration in cells that have low basal RAS activity.

2) Figure 2I and 3I should be quantified-though I believe that there is less UCP1 expression upon loss of Ace2 and more upon Ace2 overexpression, it is hard to tell visually and is certainly not obvious. Other histology stains, such as figures 5A, C, and E could also be quantified to provide additional certainty that the effect described is significant.

3) A critical issue is the lack of discussion and context in the introduction and discussion that places this work in context of the literature about RAS and energy expenditure. I recommend expanding the introduction and/or discussion to include this critical background. For example, previous work (PMID: 31638856 and 29066463) has found that RAS activation induced browning and energy expenditure which is surely relevant for the work described here, but it is not cited or discussed how these studies relate to the manuscript.

*Reviewer #2 (Recommendations for the authors):*

This is a strong paper with clear experimental evidence to support most claims put forth within the abstract (as outlined in the public review). There are a handful of points that I believe the authors should consider and/or address before final publication of this study:

1. Modifying the claims in the abstract. Most claims are well supported with evidence in the paper, as outlined in the public review. Two exceptions are:

Claim 1: "ACE2 knockout mice (ACE2-/y), Mas knockout mice (Mas-/-) and mice transplanted with brown adipose tissue from Mas-/- mice displayed impaired thermogenesis."

The authors show that wild type C57BL/6 mice transplanted with BAT from Mas-/- mice show metabolic impairments consistent with ACE2-/y and Mas-/- mice, but do not show data regarding their body temperature. This claim can be addressed by performing body temperature experiments similar to those performed in Figures 2F or 3F. Alternatively, the authors can restate their claim to state that "mice transplanted with brown adipose tissue from Mas-/- display metabolic abnormalities consistent with those seen in the ACE2 and Mas knockout mice."

Claim 2: "…impaired thermogenesis of db/db obese diabetic mice and high-fat diet-induced obese mice were ameliorated by overexpression of ACE2 or continuous fusion of Ang-(1-7)."

The authors also show that infusion of Ang-(1-7) can increase body temperature acutely after cold challenge in mice fed a high fat diet (HFD) (Figure 4H), but they do not test the effects of overexpression of ACE2 on body temperature in the HFD model. The authors can address this by either performing the body temperature experiments in ACE2-overexpressing mice in the HFD model, or reword this claim to avoid confusion over this distinction.

2. Analyzing metabolic data according to the ANCOVA method. The authors use a TSE LabMaster to gather data on the metabolic rates and physical activities of multiple mouse models in their manuscript (e.g., Figures 2C-E, Figures 3C-E, Figures 4C-E, etc.). They present the oxygen consumption and CO2 production as normalized to body weight and raised to the 0.75 power – a method previously used to report such data. However, multiple groups have raised concerns about the normalization of oxygen consumption and CO2 production to body weight, as trends regarding metabolic status can be obscured by changes in body weight. Notably, the authors find alterations in body weight in many of their models, making this a concern. Many groups now agree that analyzing metabolic data by plotting individual data points to analyze both group (i.e., phenotype) and body weight effects though ANCOVA (analysis of covariance) is the best and clearest way to present these data. See Tschop MH et al., Nat. Methods 2012 (PMID: 22205519) for more information. The authors could perform this analysis using a newly developed webtool called CalR (https://calrapp.org/) which was recently developed as a free tool for such analyses (see Mina AI et al., Cell Metab. 2018 PMID: 30017358).

---

## [Author Response]

Essential revisions:1) Body temperature experiments in ACE2-overexpressing mice in the HFD model

We have performed the body temperature experiments in ACE2-overexpressing HFD mice, which exhibited stronger thermogenesis than the control mice. This data has been integrated into the revised manuscript (Figure 3—figure supplement 1K).

2) Experiments to clarify the relationship between PKA and AKT signaling, RAS signaling and respiration

Many thanks for your suggestion. We have conducted the mitochondrial respiration in ACE2/Ang-(1-7) induced primary BATs treated with A779, PKA inhibitors, AKT inhibitors, PKA+AKT inhibitors, and the data has been integrated into the revised manuscript (Figure 7E, 7J, Figure 7—figure supplement 2C, 2F 2G, 2H).

Reviewer #1 (Recommendations for the authors):1) The authors provide ample evidence that both AKT signaling and PKA signaling contribute to RAS signaling upregulation of UCP1 and mitochondrial respiration. In their model, they have the AKT and PKA pathways as unrelated inputs into mitochondrial respiration. However, primary BATs treated with PKA and AKT inhibitors seem have a similar total loss of RAS-induced oxygen consumption. This either implies that they are either part of the same pathway (not unrelated) or that AKT and PKA inhibitors are shutting off respiration in general and that this is not related to RAS-induced respiration in particular. Some experiments to clarify the relationship between PKA and AKT signaling, RAS signaling and respiration would be helpful. For example, does co-treatment with AKT and PKA inhibitors have an additive effect?

Thanks very much for bringing out this question. Our revised data showed that co-treatment with AKT and PKA inhibitors have an additive effect in ACE2-overexpressing primary brown adipocytes cells. The data has been integrated into the revised manuscript (Figure 7—figure supplement 2G).

Do these inhibitors have an effect on respiration independent of Ang-(1-7) treatment.

We performed the treatment of AKT inhibitor (MK2206) and PKA inhibitor (H89) in primary BATs with or without of Ang-(1-7), respectively. And, we analyzed the respiration differences treated with inhibitors in different groups with or without of Ang-(1-7). Combine with the above result (Figure S6G), our replenished data indicated that Ang-(1-7) treatment induced respiration at least partly through AKT and PKA signaling pathway. The data has been integrated into the revised manuscript (Figure 7E, 7J).

It may be interesting to see if Ang-(1-7) treated cells respond differently to A779 treatment than PKA or AKT inhibition in terms of mitochondrial respiration to determine the magnitude of the effect on respiration of PKAi/AKTi are similar to the effect of a RASi.

Thanks very much for your helpful suggestion. Our fresh data showed that there is no significant difference between A779 treatment and PKA or AKT inhibitor treatment in Ang-(1-7) treated primary brown adipocytes cells. This data indicated that A779 treatment plays similar effect to PKAi/AKTi on mitochondrial respiration in Ang-(1-7) treated cells. The data has been integrated into the revised manuscript (Figure 7—figure supplement 2H).

It may be valuable to use adipocytes from the diabetic mice and induce Ace2 in those mice, then observe how AKT and PKA inhibitors effect respiration in cells that have low basal RAS activity.

Thanks very much for your valuable suggestion. After hard attempts, we found that it is difficult to get high-quality mature adipocytes from db/db mice, probably because the fat cells in db/db mice are very easy to break. However, we have still tried our best to conduct the mitochondrial respiration in ACE2 overexpressed primary brown adipocytes induced by ACE2 adenovirus infect, after treated with PKA and AKT inhibitors. Due to the RAS axis in brown adipose tissue is kind of cold induced activity, we can see a lower ACE2 or Mas level if there is no cold stimulation form figure1C-1E, which indicated the relatively low basal RAS activity. So, we performed this experiment with BATs under no cold stimulation condition. Our renewed data showed that ACE2 are similar to Ang-(1-7) in terms of mitochondrial respiration on PKAi/AKTi treatment. And, AKT inhibitors but not PKA inhibitors plays obvious effect on respiration in brown adipocytes with lower basal RAS activity. While, after induce ACE2 expression in these cells, both AKT and PKA inhibitors plays significant roles on respiration. These data also imply the contribution of AKT and PKA signal pathways to RAS activity. These data have been integrated into the revised manuscript (Figure 7—figure supplement 2C, 2F).

2) Figure 2I and 3I should be quantified-though I believe that there is less UCP1 expression upon loss of Ace2 and more upon Ace2 overexpression, it is hard to tell visually and is certainly not obvious. Other histology stains, such as figures 5A, C, and E could also be quantified to provide additional certainty that the effect described is significant.

Thanks for your concern. We have revised these data. The related Figure 2I, 3I, 4I, 5G, 6 (A, C, E) had been quantified in the revised manuscript.

3) A critical issue is the lack of discussion and context in the introduction and discussion that places this work in context of the literature about RAS and energy expenditure. I recommend expanding the introduction and/or discussion to include this critical background. For example, previous work (PMID: 31638856 and 29066463) has found that RAS activation induced browning and energy expenditure which is surely relevant for the work described here, but it is not cited or discussed how these studies relate to the manuscript.

Many thanks for bring out this valuable concern. We have added the RAS and energy expenditure related background in the introduction and corresponding discussion in the revised manuscript (line 64 p. 3, line 346 p. 14).

Reviewer #2 (Recommendations for the authors):This is a strong paper with clear experimental evidence to support most claims put forth within the abstract (as outlined in the public review). There are a handful of points that I believe the authors should consider and/or address before final publication of this study:1. Modifying the claims in the abstract. Most claims are well supported with evidence in the paper, as outlined in the public review. Two exceptions are:Claim 1: "ACE2 knockout mice (ACE2-/y), Mas knockout mice (Mas-/-) and mice transplanted with brown adipose tissue from Mas-/- mice displayed impaired thermogenesis."The authors show that wild type C57BL/6 mice transplanted with BAT from Mas-/- mice show metabolic impairments consistent with ACE2-/y and Mas-/- mice, but do not show data regarding their body temperature. This claim can be addressed by performing body temperature experiments similar to those performed in Figures 2F or 3F. Alternatively, the authors can restate their claim to state that "mice transplanted with brown adipose tissue from Mas-/- display metabolic abnormalities consistent with those seen in the ACE2 and Mas knockout mice."

Thanks very much for your helpful comments. According to your suggestion, we revised this sentence to be “mice transplanted with brown adipose tissue from Mas-/- display metabolic abnormalities consistent with those seen in the ACE2 and Mas knockout mice” (line 37 p. 2).

Claim 2: "…impaired thermogenesis of db/db obese diabetic mice and high-fat diet-induced obese mice were ameliorated by overexpression of ACE2 or continuous fusion of Ang-(1-7)."The authors also show that infusion of Ang-(1-7) can increase body temperature acutely after cold challenge in mice fed a high fat diet (HFD) (Figure 4H), but they do not test the effects of overexpression of ACE2 on body temperature in the HFD model. The authors can address this by either performing the body temperature experiments in ACE2-overexpressing mice in the HFD model, or reword this claim to avoid confusion over this distinction.

Thank you for your suggestion. We have conducted the body temperature experiments in Ace2-overexpressing HFD mice, and the data has been integrated into this version of the manuscript (Figure 3—figure supplement 1K).

2. Analyzing metabolic data according to the ANCOVA method. The authors use a TSE LabMaster to gather data on the metabolic rates and physical activities of multiple mouse models in their manuscript (e.g., Figures 2C-E, Figures 3C-E, Figures 4C-E, etc.). They present the oxygen consumption and CO2 production as normalized to body weight and raised to the 0.75 power – a method previously used to report such data. However, multiple groups have raised concerns about the normalization of oxygen consumption and CO2 production to body weight, as trends regarding metabolic status can be obscured by changes in body weight. Notably, the authors find alterations in body weight in many of their models, making this a concern. Many groups now agree that analyzing metabolic data by plotting individual data points to analyze both group (i.e., phenotype) and body weight effects though ANCOVA (analysis of covariance) is the best and clearest way to present these data. See Tschop MH et al., Nat. Methods 2012 (PMID: 22205519) for more information. The authors could perform this analysis using a newly developed webtool called CalR (https://calrapp.org/) which was recently developed as a free tool for such analyses (see Mina AI et al., Cell Metab. 2018 PMID: 30017358).

Thank you for your suggestion. We have revised the CLAMS data by ANCOVA on a webtool (https://calrapp.org/), and the data have been integrated into this version of the manuscript (Figure 2, 3, 4, 5, Figure 5—figure supplement 1).